# PHASEFORMER: FROM PATCHES TO PHASES FOR EFFICIENT AND EFFECTIVE TIME SERIES FORECASTING

**Yiming Niu[1,*] Jinliang Deng[1,2,*], Yongxin Tong[1,†]**
[1] State Key Laboratory of Complex & Critical Software Environment, Beihang University
{yimingniu, yxtong}@buaa.edu.cn
[2] Research Institute of Trustworthy Autonomous Systems, SUSTech
jinliangdeng9588@gmail.com

## ABSTRACT

Periodicity is a fundamental characteristic of time series data and has long played a central role in forecasting. Recent deep learning methods strengthen the exploitation of periodicity by treating patches as basic tokens, thereby improving predictive effectiveness. However, their efficiency remains a bottleneck due to large parameter counts and heavy computational costs. This paper provides, for the first time, a clear explanation of why patch-level processing is inherently inefficient, supported by strong evidence from real-world data. To address these limitations, we introduce a phase perspective for modeling periodicity and present an efficient yet effective solution, PhaseFormer. PhaseFormer features phase-wise prediction through compact phase embeddings and efficient cross-phase interaction enabled by a lightweight routing mechanism. Extensive experiments demonstrate that PhaseFormer achieves state-of-the-art performance on the evaluated benchmarks with around 1k parameters, consistently across benchmark datasets. Notably, it excels on large-scale and complex datasets, where models with comparable efficiency often struggle. This work marks a significant step toward truly efficient and effective time series forecasting. Code is available at this repository: https://github.com/neumyor/PhaseFormer_TSL.

## 1 INTRODUCTION

Time series forecasting underpins decision-making across diverse domains such as finance, energy, climate science, and healthcare, playing a pivotal role in tasks including weather forecasting (Qureshi et al., 2025; Wu et al., 2021a), energy consumption planning (Lai et al., 2018; Alvarez et al., 2010; Cheng et al., 2021), traffic scheduling (Cirstea et al., 2022; 2021; Wu et al., 2021b). In recent years, deep learning has demonstrated promising potential in this field by leveraging end-to-end modeling and powerful representational capacity to extrapolate from history to future trends.

A central inductive bias in forecasting models is periodicity–the recurring temporal structure inherent in many real-world time series. Periodicity are ubiquitous in practice, appearing in domains ranging from urban traffic flow to cloud resource utilization and other workloads, making periodicity-aware modeling broadly applicable (Deng et al., 2024a; Xia et al., 2025). Recent advances exploited this property by segmenting sequences into patch tokens, potentially aligned with cycles, prior to processing by the crafted models (Nie et al., 2023; Zhang & Yan, 2023; Huang et al., 2025; Tang & Zhang, 2025b). For instance, Nie et al. (2023) applied Transformer to tokenized time series to capture temporal correlations within and between cycles, while Zhang & Yan (2023) extended this paradigm by modeling cross-dimension dependencies and cross-scale interactions.

Despite their effectiveness, patch-based approaches struggle to scale efficiently to large and complex datasets (Nie et al., 2023; Zhang & Yan, 2023; Tang & Zhang, 2025b). *We attribute this poor scalability to the substantial variability of cycle patterns in real-world scenarios.* This variability stems from dynamic external factors, which continuously shift the cycle patterns. For instance,

---

*Yiming Niu and Jinliang Deng contributed equally to this work.
†Corresponding author.

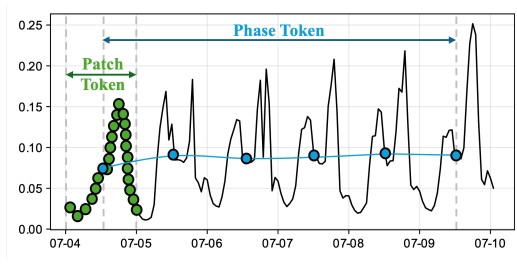

(a) Patch Token vs. Phase Token.

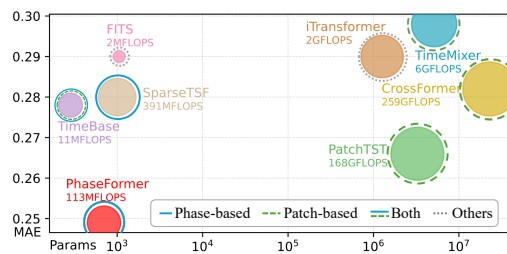

(b) Model Accuracy and Efficiency Comparison.

Figure 1: Comparison between patch-based and phase-based representations for time-series forecasting. (a) illustrates the difference in tokenization. (b) provides a joint evaluation of model accuracy, parameter scale, and computational overhead on the Traffic dataset, using an input length of 720 steps and predicting 96 future steps, where the marker size denotes the FLOPs.

traffic flow patterns may evolve as new infrastructure is introduced, while electricity demand can change with adjustments in work schedules (Shao et al., 2025b; 2026; Chengyang Zhou & Hu, 2026). This variability forces models to construct a high-dimensional representation space to faithfully accommodate the broadened distribution, which inevitably inflates both parameter counts and computational costs (Shao et al., 2025a). Additionally, these methods also struggle to generalize under such varying behavior, resulting in unreliable forecasts for samples beyond training data.

To address this challenge, we introduce a novel phase-based perspective that focuses on values aligned at the same offset across successive cycles. From this perspective, the dynamics of a time series are characterized by the cross-period trends of each phase–captured as phase tokens–while disregarding the full cyclic behavior. As illustrated in Fig. 1a, phase tokens exhibit significantly lower variability than patch tokens, enabling more efficient and generalizable representation, which in turn makes PhaseFormer well suited for resource-constrained environments. Importantly, excluding cycle patterns has minimal impact on forecasting effectiveness, since the cyclic behaviors remain locally stable and thus require little effort to predict. We study and verify these properties in depth in Sec. 3 using real-world data, showing the stationarity and compactness of the feature space offered by phase tokenization.

Building on these insights, we propose Phase-based Routing Transformer, abbreviated as **Phase-Former**, which reframes time series as a collection of phase tokens and casts step-wise prediction as phase-wise prediction. Specifically, PhaseFormer (i) aligns and extracts phase tokens from the input sequence and maps them into a shared low-dimensional latent space, (ii) employs a lightweight routing mechanism to enable efficient communication across phases, and (iii) applies a shared predictor to project the latent representations into forecasts for each phase. Extensive experiments demonstrate that, compared with PatchTST (Nie et al., 2023) and Crossformer (Zhang & Yan, 2023), PhaseFormer achieves over **99.9%** reduction in both parameter count and computational cost, while delivering consistent improvements in prediction accuracy across all seven benchmark datasets, as illustrated by the Traffic dataset in Fig. 1b. Moreover, in contrast to methods with comparable efficiency such as SparseTSF (Lin et al., 2024b) and TimeBase (Huang et al., 2025), PhaseFormer significantly enhances predictive effectiveness, particularly on large and complex datasets. Finally, we conduct a comprehensive analysis of different configurations to reveal the necessity of the constructed components and the effects of various hyperparameters. Our contributions are as follows:

1. We introduce a phase-based perspective that aligns values across cycles for the characterization of long-term time series, empirically and theoretically demonstrating improved feature stationarity and compactness over the patch-based perspective.

2. We propose PhaseFormer, a lightweight forecasting model that reframes time series as phase tokens, maps them into a shared latent space, and employs a low-rank routing mechanism specifically designed around phase tokens to enable efficient phase-wise forecasting.

3. Extensive experiments are conducted to showcase that PhaseFormer achieves substantial efficiency gains while consistently improving forecasting accuracy, establishing a superior efficiency–effectiveness trade-off across diverse benchmarks.

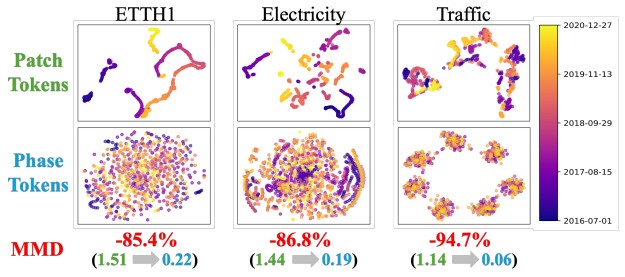 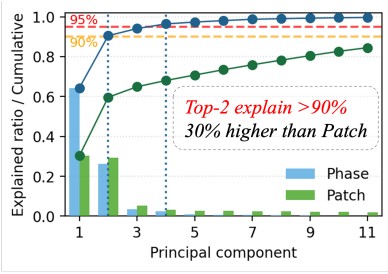

(a) Temporal shift comparison across multiple datasets.

(b) PCA for phase tokens on *Traffic*.

Figure 2: Visualization of phase tokenization and its advantages. (a) Phase tokenization yields more stable representations than patch-based embeddings. (b) Phase tokens exhibit clear low-dimensionality compared with patch tokens.

## 2 RELATED WORKS

**Transformer-Based Forecasting Architectures.** Early Transformer-based models for long sequence forecasting often overlooked the periodicity in time series (Zhou et al., 2021; Li et al., 2019). Subsequent research introduced domain-specific priors that better understand recurring temporal structures. Autoformer (Wu et al., 2021a) and FEDformer (Zhou et al., 2022) incorporated decomposition strategies and frequency-domain modeling enabling explicit representation of seasonal–trend patterns. Pyraformer (Liu et al., 2021) and Crossformer (Zhang & Yan, 2023) further enriched temporal modeling by embedding multi-scale hierarchies and cross-variable dependencies, while Liu et al. (2022) explicitly accounted for distributional shifts. More recently, PatchTST (Nie et al., 2023) reframed time series as patch sequences to enable more accurate characterization of sequence-level semantics, followed by an extension to jointly consider spatial and temporal correlations (Huo et al., 2025). Generally speaking, these models embed progressively stronger temporal biases, though often at the cost of massive parameter counts and heavy computation.

**Efficiency-Oriented Forecasting Models.** A growing body of research emphasizes efficiency, aiming to design lightweight forecasting architectures. Patch-based MLP variants such as xPatch (Stitsyuk & Choi, 2025),TimeMixer (Wang et al., 2024), and PITS (Lee et al., 2024) exploited compact tokenization or hierarchical dependencies to reduce parameter counts while maintaining accuracy. Beyond patches-based methods, frequency-based counterparts leverage spectral representations for compression and denoising. FreTS (Yi et al., 2023) applied MLPs in the frequency domain, Yi et al. (2024) learned frequency filters to improve noise robustness, and FITS (Xu et al., 2024) achieved strong accuracy with only 10k parameters. Deng et al. (2024b) demonstrated that selective decomposition can deliver both parsimony and capability. More recently, SparseTSF (Lin et al., 2024b) and TimeBase (Huang et al., 2025) highlighted the importance of cross-period correlation, sharing a similar motivation with ours. Despite their impressive computational efficiency, these methods still fall short on forecasting accuracy for large and complex datasets such as Traffic and Electricity (Lin et al., 2024b; Huang et al., 2025). Moreover, they lack systematic analysis to answer the fundamental question: *Why can phase tokens serve as an efficient alternative to patch tokens?*

## 3 MOTIVATIONS

To motivate our approach, we conduct a comparative analysis of the geometric structures of patch and phase tokens across three widely used datasets. As illustrated in Fig. 1a, a patch token is composed of adjacent observations within a local period, whereas a phase token is constructed by extracting values at identical offsets across consecutive periods. We gain the following two important insights from the thorough analysis.

**Insight 1: Phase tokens are globally stationary, while patch tokens are locally stationary.** To provide an intuitive overview of their geometric structures, we project both types of tokens into two-dimensional spaces using t-SNE (van der Maaten & Hinton, 2008). As shown in Fig. 2a, the

distributions of patch tokens drift continuously over time but exhibit local coherence, indicating *local stationarity* and supporting the minimal impact of excluding cycle patterns from intensive processing. In contrast, phase tokens form compact and coherent clusters that remain stable over the long term, reflecting strong *global stationarity*. To rigorously quantify the long-term drift, we compute the average discrepancy distance between each subsequent week and the initial week. Specifically, we adopt the Maximum Mean Discrepancy (MMD) metric (Ouyang & Key, 2021), a statistical measure of distributional divergence:

$$\text{MMD}^2(P,Q) = \mathbb{E}_{x,x'\sim P}[k(x,x')] + \mathbb{E}_{y,y'\sim Q}[k(y,y')] - 2\,\mathbb{E}_{x\sim P,y\sim Q}[k(x,y)], \quad (1)$$

where $P$ and $Q$ denote tokens collected from two different weeks, respectively, and $k(\cdot,\cdot)$ is the RBF kernel function. As two distributions become closer, their MMD value approaches zero. The results at the bottom of Fig. 2a show that the average MMD distance of the phase token space is significantly smaller than that of the patch token space. Taken together, both qualitative and quantitative analyses demonstrate that phase tokenization exhibits substantially lower temporal distribution divergence, thereby *facilitating better generalization across the time axis*.

**Insight 2: Phase tokens reside in a lower-dimensional subspace than patch tokens.** To measure the effective dimensionality of the token space, we perform principal component analysis (PCA) on it. Surprisingly, as illustrated in Fig. 2b, two dimensions are already sufficient to explain over 90% of the variance of phase tokens, whereas patch tokens require more than eleven dimensions to achieve the same degree of explanation, owing to their drifting behavior observed in Fig. 2a. Consequently, phase information resides in a low-dimensional subspace, *providing a principled basis for parameter- and computation-efficient modeling*.

We further establish, using perturbation theory, that phase tokenization remains stable under perturbations of cycle patterns, whereas patch tokenization exhibits structural drift. To formalize this result, we represent a univariate periodic sequence as a two-dimensional matrix via delay embedding and analyze the structural properties of this embedded matrix. Due to space limitations, only the core theorem is presented here, while the detailed proof is provided in Sec. A.7.

**Theorem 1 (Phase Tokenization Stability)** *Let* $X = AG^\top + N \in \mathbb{R}^{D \times H}$ *with* $\text{rank}(A) = \text{rank}(G) = r \ll \min(D, H)$, *and consider the transformed data*

$$X' = XS^\top + R, \quad (2)$$

*where* $\|N'\|_2 \le \|S\|_2\|N\|_2$, $\|R\|_2 \le \varepsilon(\|M\|_F + \|N\|_F)$, *and let* $\delta_{\min} > 0$ *denote the minimal spectral separation. Then there exists a universal constant* $C > 0$ *such that:*

1. *For phase tokenization and corresponding subspace* $\mathcal{U}_r$, *there exists:*

$$d\big(\mathcal{U}_r(X), \mathcal{U}_r(X')\big) \le C \frac{\|N\|_2 + \|N'\|_2 + \|R\|_2}{\delta_{\min}}, \quad (3)$$

   *with exact invariance in the noiseless case* $(N = R = 0)$.

2. *For patch tokenization and corresponding subspace* $\mathcal{V}_r$, *there exists:*

$$d\big(\mathcal{V}_r(X), \mathcal{V}_r(X')\big) \ge d\big(\text{Col}(G), \text{Col}(SG)\big) - C \frac{\|N\|_2 + \|N'\|_2 + \|R\|_2}{\delta_{\min}}. \quad (4)$$

**Takeaways.** Phase tokenization is structurally invariant under the cycle pattern change $S$ and only subject to perturbations from noise and small day-to-day mismatches. In contrast, patch tokenization generally suffers from a non-vanishing structural offset. Hence, *phase tokenization is more robust and consistent under cycle pattern drifts*.

## 4 METHODOLOGY

Given the focus on periodicity, we adopt the channel-independent paradigm (Nie et al., 2023; Zeng et al., 2023) and omit the channel dimension throughout the remainder of this paper. The objective of forecasting is to predict the future trajectory $\mathbf{Y} \in \mathbb{R}^{L_{\text{out}}}$ from an input sequence $\mathbf{X} \in \mathbb{R}^{L_{\text{in}}}$, where $L_{\text{in}}$ and $L_{\text{out}}$ denote the input and output lengths, respectively. In the following sections, we describe the data preprocessing procedure, present the proposed network architecture, and finally analyze the computational complexity of the method.

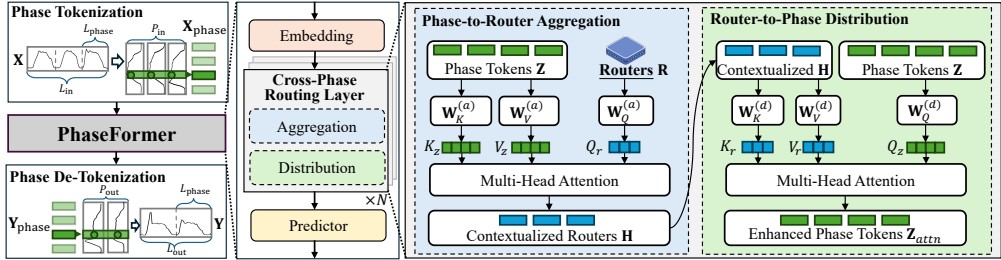

Figure 3: The overview of PhaseFormer.

## 4.1 DATA PRE-PROCESSING

**Normalization and De-Normalization.** Following Kim et al. (2021), we normalize inputs with their estimated mean and standard deviation, and de-normalize predictions to the original scale.

**Phase Tokenization and De-Tokenization.** Phase tokenization transforms the one-dimensional input sequence into a two-dimensional phase–period matrix for the following processing. Conversely, phase de-tokenization reconstructs the predicted phase–period matrix back into a one-dimensional output sequence. Let $L_{\text{phase}}$ denotes the period length, which can be estimated using frequency analysis as we further illustrated in Sec. 5.3, and remain fixed during the process.

To ensure that the input sequence length is a multiple of $L_{\text{phase}}$, we circularly pad the sequence to length $P_{\text{in}} * L_{\text{phase}}$, where $P_{\text{in}} = \left\lceil \frac{L_{\text{in}}}{L_{\text{phase}}} \right\rceil$. As illustrated in Fig. 3, the padded sequence $\mathbf{X}$ is then reshaped into a phase–period matrix $\mathbf{X}_{\text{phase}} \in \mathbb{R}^{L_{\text{phase}} \times P_{\text{in}}}$, where each entry $\mathbf{X}_{\text{phase}}[\ell, p]$ corresponds to the observation at the $\ell^{\text{th}}$ phase of the $p^{\text{th}}$ period. In the de-tokenization process, the predicted phase–period matrix is mapped back to the temporal domain by reversing the transformation, thereby reconstructing the final one-dimensional forecast sequence.

## 4.2 PHASE-BASED ROUTING TRANSFORMER

The phase–period matrix is fed into our proposed phase-based routing Transformer, termed Phase-Former, to capture and extrapolate temporal dynamics at the phase level in an efficient and effective way. As illustrated in Fig. 3, PhaseFormer first applies an embedding layer to the phase tokens, then refines them through multiple cross-phase routing layers, and finally maps them to the target via a shared predictor. Next, we elaborate on the design of these modules in detail.

### 4.2.1 EMBEDDING LAYER

The embedding layer projects the phase tokens $\mathbf{X}_{\text{phase}}$ into a low-dimensional representation space, allowing the informative components to be extracted from raw observations that are often contaminated by perturbations. Formally, for each phase index $\ell \in \{1, \ldots, L_{\text{phase}}\}$, the corresponding phase token $\mathbf{X}_{\text{phase}}[\ell, :]$ is mapped into a $d$-dimensional representation through a linear function $f_\theta$, parameterized by $\theta \in \mathbb{R}^{P_{\text{in}} \times d}$:

$$\mathbf{Z} = f_\theta(\mathbf{X}_{\text{phase}}) \in \mathbb{R}^{L_{\text{phase}} \times d} \tag{5}$$

To better capture the temporal ordering among phases, we introduce a set of learnable positional embeddings $\mathbf{E}_{\text{pos}} \in \mathbb{R}^{L_{\text{phase}} \times d}$ to distinguish the relative position of each phase, following Liu et al. (2023a). These embeddings are added to $\mathbf{Z}$ in a phase-wise manner, so that each phase representation is enriched with its positional information:

$$\tilde{\mathbf{Z}} = \mathbf{Z} + \mathbf{E}_{\text{pos}}. \tag{6}$$

The resulting $\tilde{\mathbf{Z}}$ is then forwarded to the cross-phase routing layers for higher-level feature interaction and forecasting.

### 4.2.2 Cross-Phase Routing Layer

Directly modeling full pairwise interactions among phase representations via self-attention is computationally expensive. To handle this, we introduce a set of learnable routers $\mathbf{R} \in \mathbb{R}^{M \times d}$ to mediate information exchange across phases. While the design is inspired by prior routing-style design (Jaegle et al., 2021; Zhang & Yan, 2023), our contribution lies in *leveraging the inherent low-rank structure of phase-aligned tokens*. The cross-phase routing substantially reducing the quadratic cost of self-attention while preserving rich cross-phase dependencies.

Cross-phase routing consists of two steps: (i) *phase-to-router aggregation*, which selectively compresses information from phase representations into the compact set of routers; and (ii) *router-to-phase distribution*, which selectively propagates the aggregated cross-phase information from the routers back to the phase representations. Both steps are implemented via cross-attention, allowing the model to scale efficiently while preserving strong representational capacity.

**Phase-to-Router Aggregation.** The routers attend to the phase representations to extract contextual information, yielding contextualized router embeddings $\mathbf{H} \in \mathbb{R}^{M \times d}$. Specifically, the routers act as queries while the phases provide keys and values. The projection matrices $\mathbf{W}_Q^{\mathrm{agg}}, \mathbf{W}_K^{\mathrm{agg}}, \mathbf{W}_V^{\mathrm{agg}} \in \mathbb{R}^{d \times d}$ map the representations into query, key, and value spaces, respectively:

$$\mathbf{Q}_r = \mathbf{R}\mathbf{W}_Q^{\mathrm{agg}}, \quad \mathbf{K}_z = \tilde{\mathbf{Z}}\mathbf{W}_K^{\mathrm{agg}}, \quad \mathbf{V}_z = \tilde{\mathbf{Z}}\mathbf{W}_V^{\mathrm{agg}}. \tag{7}$$

The aggregated router embeddings are then obtained via multi-head attention (MHA) with $d_h$ heads:

$$\mathbf{H} = \mathrm{MHA}(\mathbf{Q}_r, \mathbf{K}_z, \mathbf{V}_z). \tag{8}$$

**Router-to-Phase Distribution.** The aggregated information in the routers is subsequently redistributed to the phase representations, thereby enabling cross-phase information flow. In this step, the phase representations serve as queries while the routers provide keys and values, yielding refined phase representations $\mathbf{Z}_{\mathrm{attn}}$. The projection matrices $\mathbf{W}_Q^{\mathrm{dist}}, \mathbf{W}_K^{\mathrm{dist}}, \mathbf{W}_V^{\mathrm{dist}} \in \mathbb{R}^{d \times d}$ are used for this distribution:

$$\mathbf{Q}_z = \tilde{\mathbf{Z}}\mathbf{W}_Q^{\mathrm{dist}}, \quad \mathbf{K}_r = \mathbf{H}\mathbf{W}_K^{\mathrm{dist}}, \quad \mathbf{V}_r = \mathbf{H}\mathbf{W}_V^{\mathrm{dist}}, \tag{9}$$

$$\mathbf{Z}_{\mathrm{attn}} = \mathrm{MHA}(\mathbf{Q}_z, \mathbf{K}_r, \mathbf{V}_r). \tag{10}$$

This mechanism restores phase-level resolution while simultaneously enforcing coherence across phases through the contextualized routers. Ultimately, each phase representation attends to all others through a two-stage routing pathway.

### 4.2.3 Predictor

The predictor produces multi-step forecasts of length $P_{\mathrm{out}}$ for all phases simultaneously, based on their refined representations. Taking as input the refined phase representations $\mathbf{Z}_{\mathrm{attn}} \in \mathbb{R}^{L_{\mathrm{phase}} \times d}$ from the final cross-phase routing layer, the predictor is realized as a linear mapping $g_\phi$, parameterized by $\phi \in \mathbb{R}^{d \times P_{\mathrm{out}}}$:

$$\mathbf{Y}_{\mathrm{phase}} = g_\phi(\mathbf{Z}_{\mathrm{attn}}) \in \mathbb{R}^{L_{\mathrm{phase}} \times P_{\mathrm{out}}}. \tag{11}$$

All phases share the same predictor parameters, which enforces consistency across phases and reduces the number of trainable parameters. This not only improves efficiency but also regularizes learning, thereby enhancing generalization. Finally, the predicted phase–period matrix $\mathbf{Y}_{\mathrm{phase}}$ is passed through de-tokenization and de-normalization to produce the final forecast $\mathbf{Y}$.

### 4.3 Complexity of PhaseFormer

For each variable, the overall complexity of PhaseFormer can be summarized as follows: the phase embedding layer requires $O(L_{\mathrm{phase}} P_{\mathrm{in}} d)$ time and $O(L_{\mathrm{phase}} d)$ memory. The cross-phase routing layer, which dominates computation, incurs $O((L_{\mathrm{phase}}+M)d^2 + M L_{\mathrm{phase}} d)$ time and $O(HML_{\mathrm{phase}} + (L_{\mathrm{phase}}+M)d)$ memory. Finally, the predictor costs $O(L_{\mathrm{phase}} d P_{\mathrm{out}})$ time and $O(L_{\mathrm{phase}} P_{\mathrm{out}})$ memory. Aggregating across $N$ blocks, the end-to-end time complexity is:

$$O\Big(N\big((L_{\mathrm{phase}} + M)d^2 + M L_{\mathrm{phase}} d\big) + L_{\mathrm{phase}} d(P_{\mathrm{in}} + P_{\mathrm{out}})\Big).$$

Substituting $P_{\text{in}} = \lceil L_{\text{in}}/L_{\text{phase}} \rceil$ and $P_{\text{out}} = \lceil L_{\text{out}}/L_{\text{phase}} \rceil$ into the above expression gives:

$$O\Big( N\big((L_{\text{phase}} + M)d^2 + ML_{\text{phase}}d\big) + d(L_{\text{in}} + L_{\text{out}}) \Big).$$

As investigated in Sec. 3, the phase token space exhibits a inherently low-dimensional structure, which allows $M$ and $d$ to be chosen as fixed and small numbers. Thus, the computational cost grows in a linear manner with both the input length $L_{\text{in}}$ and the output horizon $L_{\text{out}}$.

### 4.4 DISCUSSION

**Discussion on Periodic Alignment.** Although PhaseFormer is built around a periodic alignment mechanism, it does not completely fail on weakly periodic or partially non-periodic time series. When no strong periodic structure is present, the phase representation naturally degenerates into a coarse-scale subsampling of the sequence. These coarse phase tokens function similarly to trend embeddings sampled every k steps, capturing slowly varying components and broad temporal structures. Through the routing mechanism, the model can still aggregate information across these coarse tokens, thereby suppressing high-frequency noise and emphasizing the underlying smooth temporal evolution. This prevents overfitting to short-term stochastic fluctuations and enables the model to retain a certain degree of robustness even without clear periodicity.

**Discussion on Failure Modes.** In datasets lacking a clear dominant period or exhibiting only weak periodicity, the automatically estimated phase length may be affected by noise or minor harmonics, resulting in unstable or incorrect phase alignment. Such misalignment constitutes a primary failure mode for the model. In multi-period settings, if the model aligns to a suboptimal harmonic instead of the true dominant component, it can still operate but tends to deliver inferior predictions, as we later shown in Sec. 5.3.

**Discussion on Padding.** PhaseFormer employs a cycle-padding mechanism to ensure alignment along the phase dimension. However, when the chosen phase length is excessively large or small, a substantial amount of padding is introduced. Excessive padding may inject artificial boundary information into the input, creating boundary artifacts that degrade prediction quality. Selecting an appropriate phase length is therefore crucial for minimizing padding and maintaining stable performance. In practice, using input sequence lengths that are divisible by the phase length is recommended whenever possible to avoid unnecessary padding and its associated artifacts.

## 5 EXPERIMENTS

### 5.1 LONG-TERM TIME SERIES FORECASTING

We conduct a joint evaluation of model efficiency and predictive accuracy. The comparative analysis highlights that the proposed *PhaseFormer* establishes an improved effectiveness-efficiency tradeoff in terms of parameter scale and error metrics. We also provide the code in `https://github.com/neumyor/PhaseFormer_TSL`.

**Datasets and Setup.** Experiments are performed on seven widely used long-term time series forecasting datasets: *ETTh1*, *ETTh2*, *ETTm1*, *ETTm2*[1], *Weather*[2], *Electricity*[3], and *Traffic*[4], covering a diverse range of real-world scenarios. The details of the datasets are provided in Sec. A.2. Following prior works (Nie et al., 2023; Zhang & Yan, 2023; Huang et al., 2025), we adopt a 6:2:2 split for the ETT datasets and a 7:1:2 split for the other datasets. For PhaseFormer, we report the average results over three random seeds, while for the other baselines we follow their official implementations and released code. We evaluate the forecasting **accuracy** of all tested models using mean squared error (MSE) and mean absolute error (MAE), and assess **efficiency** in terms of floating-point operations (FLOPs) and the number of parameters (Params).

---

[1]`https://github.com/zhouhaoyi/ETDataset`
[2]`https://www.bgc-jena.mpg.de/wetter/`
[3]`https://archive.ics.uci.edu/ml/datasets`
[4]`https://pems.dot.ca.gov/`

Table 1: Main results for long-term forecasting. The input sequence length is $L_{\text{input}} = 720$, and results are averaged over forecast horizons $L_{\text{out}} \in \{96, 192, 336, 720\}$. The best results are shown in **bold**, and the second-best in underline.

| Dataset | PhaseFormer | | PatchTST | | iTransformer | | Crossformer | | FEDformer | | TimeBase | | SparseTSF | | FITS | | TimeMixer | |
|---|---|---|---|---|---|---|---|---|---|---|---|---|---|---|---|---|---|---|
| | MSE | MAE | MSE | MAE | MSE | MAE | MSE | MAE | MSE | MAE | MSE | MAE | MSE | MAE | MSE | MAE | MSE | MAE |
| ETTh1 | **0.403** | **0.415** | 0.420 | 0.439 | 0.453 | 0.467 | 0.517 | 0.512 | 0.523 | 0.523 | 0.404 | 0.416 | 0.406 | 0.418 | 0.419 | 0.435 | 0.452 | 0.474 |
| ETTh2 | 0.346 | 0.388 | 0.344 | 0.390 | 0.392 | 0.422 | 1.468 | 0.867 | 0.428 | 0.469 | 0.347 | 0.397 | 0.345 | 0.383 | **0.334** | **0.382** | 0.386 | 0.425 |
| ETTm1 | **0.346** | **0.374** | 0.354 | 0.383 | 0.370 | 0.401 | 0.390 | 0.417 | 0.438 | 0.465 | 0.356 | 0.380 | 0.362 | 0.383 | 0.359 | 0.382 | 0.383 | 0.413 |
| ETTm2 | **0.250** | **0.313** | 0.251 | 0.319 | 0.278 | 0.337 | 0.392 | 0.426 | 0.401 | 0.452 | **0.250** | 0.314 | 0.252 | 0.316 | 0.285 | 0.336 | 0.314 | 0.367 |
| Electricity | **0.160** | **0.250** | 0.169 | 0.265 | 0.165 | 0.263 | 0.180 | 0.273 | 0.235 | 0.348 | 0.167 | 0.258 | 0.168 | 0.263 | 0.172 | 0.270 | 0.171 | 0.273 |
| Traffic | **0.386** | **0.249** | 0.394 | 0.266 | 0.406 | 0.290 | 0.545 | 0.282 | 0.638 | 0.400 | 0.418 | 0.278 | 0.413 | 0.280 | 0.410 | 0.290 | 0.421 | 0.298 |
| Weather | **0.223** | **0.260** | 0.223 | 0.264 | 0.233 | 0.273 | 0.255 | 0.304 | 0.354 | 0.393 | 0.227 | 0.262 | 0.243 | 0.285 | 0.241 | 0.283 | 0.237 | 0.281 |

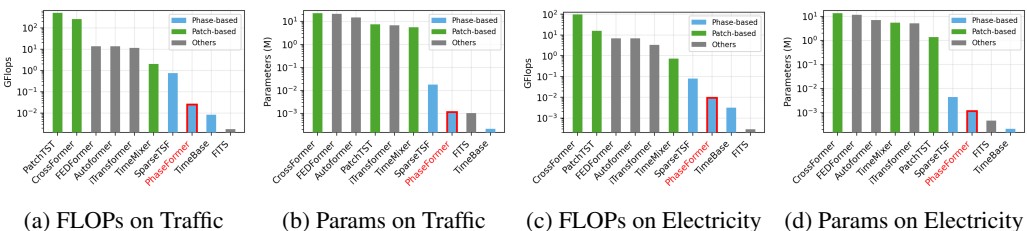

(a) FLOPs on Traffic    (b) Params on Traffic    (c) FLOPs on Electricity    (d) Params on Electricity

Figure 4: Comparison of FLOPs and parameter counts across models on the Traffic and Electricity, using an input length of 720 steps and predicting 96 future steps. Patch-based models are shown in green, phase-based models in blue, and other models in gray.

**Baselines and Implementation Details.** We evaluate our approach against eight competitive baselines, following the standard non-pretraining setting where all models are trained and evaluated directly on domain-specific datasets. The baselines span both state-of-the-art Transformer-based architectures and recent parameter-efficient forecasting models. We compare our method with PatchTST(2023), iTransformer(2023b), Crossformer(2023), FEDformer(2022), TimeBase(2025), SparseTSF(2024b), FITS(2024), and TimeMixer(2024). Among these, PatchTST, Crossformer, and TimeMixer are patch-based; SparseTSF is phase-based; TimeBase integrates patch and phase paradigms; FITS and FEDformer are frequency-domain; and iTransformer models the full sequence directly. For all baselines, we adopt the recommended configurations provided in their official implementations, keeping hyperparameters aligned with Huang et al. (2025). The model is optimized using the Adam optimizer with a fixed learning rate of $1 \times 10^{-3}$. Following the settings from efficiency-oriented works (Huang et al., 2025; Lin et al., 2024b; Xu et al., 2024), the look-back length is set to 720 time steps. More implementation details are provided in Sec. A.2.

**Main Results.** We evaluate the predictive accuracy of PhaseFormer and the baseline methods on seven datasets. Tab. 1 reports the average prediction errors across four forecasting horizons, with detailed results provided in Sec. A.3.1. Overall, PhaseFormer consistently achieves superior performance on nearly all datasets, with particularly notable gains on complex and dynamic benchmarks such as Weather, Electricity, and Traffic. For example, on the largest dataset, Traffic, PhaseFormer surpasses the second-best method, PatchTST, by 6.3% and outperforms TimeBase by 10.4%, underscoring its robustness on large-scale and heterogeneous data. The only exception is ETTh2, where PhaseFormer ranks second to FITS while still maintaining highly competitive accuracy. A closer examination reveals that patch-based baselines, including PatchTST, Crossformer, and TimeMixer, exhibit performance degradation on the Electricity, likely due to stronger distributional shifts.

**Efficiency Comparison.** We evaluate the computational overhead of all models, with detailed results in Sec. A.3.2. Fig. 4 shows the FLOPs and the number of parameters of all tested models on the Electricity and Traffic. Overall, phase-based models incur lower overhead than patch-based ones. On the Traffic dataset, PhaseFormer achieves an extraordinary FLOPs reduction of about 99.99% over PatchTST and Crossformer. Beyond patch-based baselines, it also outperforms other phase-based models like SparseTSF, consistently delivering high efficiency. This stems from the lower variety of phase tokens over time (Sec. 3), making them inherently more efficient to process. Taken together with the previous accuracy evaluations, these results clearly demonstrate that PhaseFormer provides an efficient yet effective solution, delivering superior performance on complex datasets.

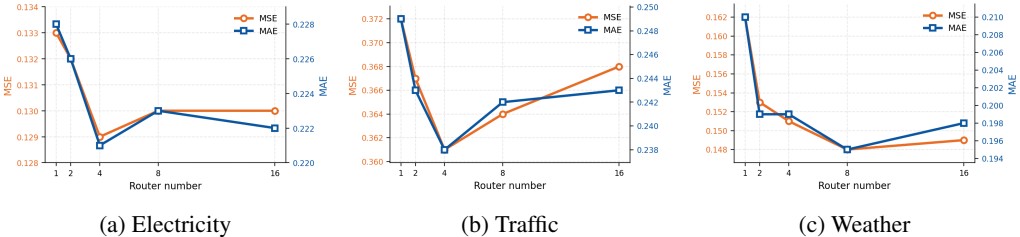

(a) Electricity        (b) Traffic        (c) Weather

Figure 5: Effect of varying the number of routers $M$ on forecasting performance on three datasets.

Table 2: Cross-Phase Routing layer ablation. Each cell reports MSE, MAE, and FLOPs. Lower is better for all metrics. FLOPs are reported in millions (MFLOPs). The best results are highlighted with **Bold**, and the second-best results with Underlined.

| Dataset | PhaseFormer | | | w/ FullAttention | | | w/ LinearMixing | | | w/o Routing | | |
|---|---|---|---|---|---|---|---|---|---|---|---|---|
| | MSE | MAE | FLOPs | MSE | MAE | FLOPs | MSE | MAE | FLOPs | MSE | MAE | FLOPs |
| Weather | **0.1503** | **0.1971** | 3.119 | 0.1527 | 0.2005 | 3.202 | 0.1700 | 0.2226 | 0.920 | 0.1907 | 0.2406 | **0.783** |
| Electricity | **0.1290** | **0.2209** | 42.213 | 0.1295 | 0.2217 | 48.951 | 0.1403 | 0.2334 | 14.068 | 0.1423 | 0.2365 | **11.972** |
| Traffic | **0.3721** | **0.2475** | 113.356 | 0.3791 | 0.2513 | 131.452 | 0.3842 | 0.2532 | 37.776 | 0.3892 | 0.2584 | **32.149** |

## 5.2 ABLATION STUDIES AND ANALYSIS

**Varying the Number of Routers.** We systematically evaluate the impact of different number of routers $M$ on model performance, with results summarized in Fig. 5. The experiments indicate that across three datasets, the model's prediction error generally decreases as the number of routers $M$ increases, before eventually stabilizing or slightly rising. It is worth noting that the best performance is usually achieved when $M \in \{4, 8\}$, which is much smaller compared to the actual number of phase tokens, $L_{\text{phase}} = 24$. This observation indicates that the phase token spans an inherently low-dimensional space, so only a small number of routers is sufficient to effectively capture and represent its underlying structure. More detailed results are provided in Sec. A.3.4.

**Effectiveness of Cross-Phase Routing.** To assess the contribution of the cross-phase routing layer, we compare four variants of the model: **PhaseFormer**, which adopts the original cross-phase routing layer; **w/ FullAttention**, which substitutes the cross-phase routing layer with a full attention mechanism; and **w/ LinearMixing**, which replaces the cross-phase routing layer with a linear layer; and **w/o Routing**, which directly projects each phase into its own future. All other experimental settings are kept identical across these variants.

As summarized in Tab. 2, PhaseFormer consistently outperforms **w/ LinearMixing** and **w/o Routing**, indicating that explicit cross-phase routing is crucial for modeling periodic dynamics. Moreover, PhaseFormer not only incurs less computational and memory overhead, but also achieves lower prediction error than **w/ FullAttention**, showing that the routing layer is both efficient and effective. We attribute these gains to operating in a low-dimensional phase token space, which concentrates informative interactions and reduces cost.

## 5.3 SENSITIVITY ANALYSIS OF PHASE LENGTH CHOICES

sysname automatically determines the phase length through frequency-domain analysis (see Sec. A.2), extracting salient periodic components directly from the data. To assess its sensitivity to phase length, we conduct experiments on the Traffic dataset. Its frequency spectrum contains several clear periodic components (see Fig. 6). Based on spectral amplitudes, we select the top five candidate phase lengths and run PhaseFormer for forecasting. The results are shown in Tab. 3.

| Phase Length | MSE | MAE |
|---|---|---|
| 24 | 0.3619 | 0.2384 |
| 12 | 0.3960 | 0.2765 |
| 8 | 0.4032 | 0.2801 |
| 28 | 0.4063 | 0.2753 |
| 21 | 0.4184 | 0.2970 |

Table 3: Comparison of forecasting performance under different phase lengths on the Traffic dataset (lookback=720, horizon=96).

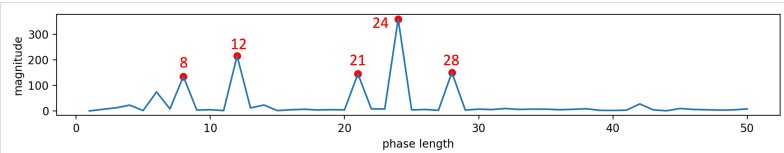

Figure 6: The spectrum calculated on the Traffic dataset. The top-5 phase length is 24, 12, 8, 28, 21.

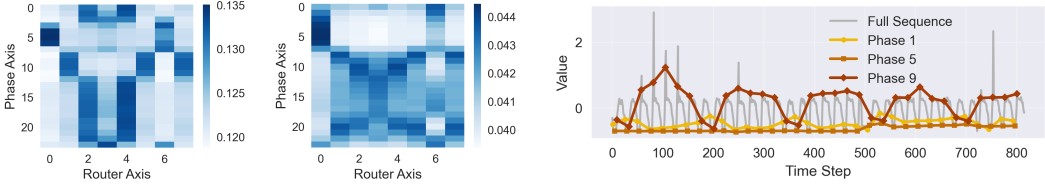

(a) Aggregation Weights     (b) Distribution Weights     (c) Visualization of three selected phase tokens

Figure 7: Case study on a sample from the Traffic dataset. (a) Attention weight matrix during Phase-to-Router aggregation. (b) Attention weight matrix during Router-to-Phase distribution. Both matrices capture the association between 8 routers and 24 input phases. (c) Visualization of three representative phases (1, 5, and 9), each represents a distinct attentive pattern with routers.

The experiments show that the model performs best when the phase length is 24, corresponding to the dominant spectral component. As the phase length deviates from this main period, forecasting errors increase, consistent with the magnitude patterns in the frequency domain. These findings demonstrate that PhaseFormer effectively captures the dataset's primary periodic structure.

## 5.4 CASE STUDY

We select one sample from Traffic dataset, comprising an input sequence of length 720 and an output sequence of length 96 (816 time steps in total). The input sequence is fed into PhaseFormer, and we record the attention-weight matrices at the first cross-phase routing layer during both aggregation (Phase→Router) and distribution (Router→Phase). As shown in Fig. 7a and Fig. 7b, both attention patterns exhibit clear local similarity: adjacent phases tend to be assigned to the same routers and to receive attention from similar routers. This indicates that the routing mechanism captures temporally consistent phase relationships. Meanwhile, the attention weights reveal that certain phases share similar attentive patterns. To analyze this further, we focus on three phases with distinct attentive patterns and visualized them in Fig. 7c. These phases display different temporal behaviors: Phase 5 remains relatively stable over long horizons, whereas Phase 9 and Phase 1 both exhibit a pronounced 7-day periodicity but with opposite trends. The differing patterns of these phase tokens suggest that the router structure not only distinguishes among phase tokens but also effectively models their periodicity and trend characteristics.

## 6 CONCLUSION

This work identifies the inefficiencies of patch-based forecasting and presents PhaseFormer, a phase-centric model that captures periodicity via compact phase representations and lightweight cross-phase routing. Both theoretical analysis and empirical validation converge on the same conclusion that phase representations remain both more robust and more efficient than patch-based approaches under cycle pattern shifts. Consequently, PhaseFormer maintains high predictive accuracy while remaining lightweight compared to patch-based methods. More broadly, these results provide a practical pathway for building lightweight yet powerful forecasting models that retain accuracy without heavy and complex architectures.

However, the approach assumes locally stable periodicity across the input and output horizons; under highly irregular or non-repetitive cycles, phase representations may fail to capture meaningful dynamics. Future work will relax this assumption by modeling non-stationarity and complex drifts, aiming to develop more resilient phase representations and further establish PhaseFormer as a benchmark for long-term time-series forecasting.

## 7 ETHICS STATEMENT

This study focuses on methodological advances in time-series forecasting and does not involve human subjects, personally identifiable data, or sensitive private information. All experiments use publicly available benchmark datasets that are widely adopted in the research community, and their use complies with the terms of release. We do not employ proprietary or confidential data, and no conflicts of interest exist. The contributions are purely technical in nature and do not promote harmful applications. All authors affirm adherence to fairness, research integrity, and relevant legal and ethical standards, in line with the ICLR Code of Ethics.

## 8 REPRODUCIBILITY STATEMENT

We make substantial efforts to ensure reproducibility. All datasets used in our experiments are publicly accessible, with links provided in Sec. 5. Detailed dataset statistics, preprocessing steps, and partitioning procedures appear in Sec. A.2. Model architectures, hyperparameters, and training procedures (including optimizer choice, learning rate, look-back window length, and router configuration) are described in Sec. 5 and the Sec. A.2.

For fair comparison, we follow the official implementations of all baseline models and provide references to their sources. Comprehensive experimental results, including ablation studies, efficiency analyses, and visualizations, appear in Sec. 5 and Sec. A.3. Theoretical analyses supporting our design choices also appear in the Sec. A.7.

Finally, to facilitate independent verification, we release anonymized source code and experiment scripts in a public repository at `https://github.com/neumyor/PhaseFormer_TSL`. Collectively, these measures ensure that our reported results are reliably reproducible and extensible by the research community.

## 9 ACKNOWLEDGMENT

The authors would like to thank all the anonymous reviewers for their insightful comments. This work was partially supported by National Science Foundation of China (NSFC) (Grant Nos. 62425202, 62336003), the Beijing Natural Science Foundation (Z230001), the Fundamental Research Funds for the Central Universities No. JK2024-03, the Didi Collaborative Research Program, the State Key Laboratory of Complex & Critical Software Environment (SKLCCSE), the Open Project Program of State Key Laboratory of Virtual Reality Technology and Systems, Beihang University (No.VRLAB2024A02).

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

# A APPENDIX

## A.1 DETAILS ABOUT BASELINES

In our experiments, we incorporated a diverse set of time series forecasting models, with particular emphasis on approaches based on **Patch Tokenization** and efficient forecasting models. The details of these models are as follows:

1. **PatchTST** — A channel-independent Transformer that treats each variable as an individual channel and segments the time series into patches as tokens. This design reduces the complexity of the attention mechanism and enables the utilization of longer historical sequences, thereby improving long-term forecasting accuracy.

2. **iTransformer** — A channel-dependent Transformer that models variables themselves as tokens to capture inter-variable relationships, while simultaneously accounting for nonlinear temporal variations within each variable.

3. **Crossformer** — A multi-scale Transformer that performs patching or segmentation along the temporal dimension and employs a two-stage attention mechanism (within-time and cross-variable). This design effectively captures both temporal dependencies and inter-variable correlations, making it particularly suitable for datasets characterized by strong inter-variable coupling and mixed long- and short-term patterns.

4. **FEDformer** — A model that integrates trend-seasonal decomposition with frequency-domain analysis. It extracts a small number of significant frequency components to enhance periodic forecasting performance while maintaining controlled model complexity in long-term forecasting tasks.

5. **SparseTSF** — A lightweight model that reduces temporal complexity through periodic down-sampling or subsequence selection, aiming to achieve competitive periodic forecasting performance with minimal resource consumption.

6. **FITS** — A lightweight model that leverages frequency-domain features and interpolation operations to reconstruct the predicted sequences. With fewer parameters and low computational overhead compared with other models, it demonstrates strong performance on time series with distinct spectral structures.

7. **TimeBase** — A model that constructs temporal bases (via patching or segmentation strategies) to represent historical and future variations. Its objective is to maintain satisfactory forecasting accuracy while reducing computational and parameter costs.

8. **TimeMixer** — An patch-based forecasting model fully based on MLPs. It employs Past-Decomposable-Mixing to decouple seasonal and trend components across different scales (fine and coarse) and utilizes Future-Multipredictor-Mixing to aggregate multi-scale predictions. This design achieves a balance of efficiency and accuracy in both short-term and long-term forecasting tasks.

9. **CycleNet(2024a)** — A lightweight model explicitly modeling stable periodicity to enhance the performance of models in long-term time series forecasting. It introduces the Residual Cycle Forecasting (RCF) technique, which utilizes learnable recurrent cycles to model the inherent periodic patterns within sequences, and then performs predictions on the residual components of the modeled cycles.

10. **PatchMLP(2025a)** — A Patch-based MLP model. It leverages a patching mechanism to enhance sequence locality and replaces complex Transformer architectures with a simple yet highly effective MLP framework.

## A.2 IMPLEMENTATION DETAILS

We present detailed statistics of the datasets in Tab. 4. The data loading and preprocessing procedures follow prior works (Nie et al., 2023; Huang et al., 2025).

All baseline methods are implemented based on their original papers or official code. For cases where fixed random seeds are not specified, each experiment is repeated three times to ensure stability. All experiments are conducted using PyTorch (Paszke et al., 2019) on a single NVIDIA A100 24GB GPU.

| Dataset | Var | Length | $T$ | $L$ | Freq | Scale |
|---|---|---|---|---|---|---|
| ETTh1 | 7 | 14,400 | 720 | 96~720 | 1hour | 0.1M |
| ETTh2 | 7 | 14,400 | 720 | 96~720 | 1hour | 0.1M |
| ETTm1 | 7 | 57,600 | 720 | 96~720 | 15mins | 0.4M |
| ETTm2 | 7 | 57,600 | 720 | 96~720 | 15mins | 0.4M |
| Weather | 21 | 52,696 | 720 | 96~720 | 10mins | 1.1M |
| Electricity | 321 | 26,304 | 720 | 96~720 | 1hour | 8.1M |
| Traffic | 862 | 17,544 | 720 | 96~720 | 1hour | 15.0M |

Table 4: Dataset statistics used in experiments.

For model configuration, the primary period is obtained from frequency-domain analysis by identifying the dominant spectral component and remains fixed throughout training. The number of routers is selected via grid search on validation set from 1 to 8, and the inner dimension of Phase-Former is tuned over $\{8, 16, 32, 64, 128\}$ using the same procedure. The depth of the model is selected from $\{1, 2, 3\}$ using the same grid-search procedure, while the number of attention heads is tuned over $\{1, 4, 8\}$. For other hyperparameters such as learning rate, batch size, optimizer settings, and early-stopping patience, we adopt commonly used configurations in the current research community, consistent with prior work (Huang et al., 2025). Please refer to the released code for complete training details at `https://github.com/neumyor/PhaseFormer_TSL`.

## A.3 FULL RESULTS

### A.3.1 THE DETAILED FORECASTING ACCURACY RESULTS

We present detailed forecasting results across all prediction horizons on the test sets in Tab. 6, with the input length fixed to 720. PhaseFormer consistently delivers strong and stable performance across most datasets and forecasting lengths. Starting from the general observations, PhaseFormer shows clear advantages in a wide range of real-world scenarios. In particular, when compared with CycleNet (an advanced model designed for periodic structures), PhaseFormer achieves 28 Top-1 results versus CycleNet's 15. This substantial difference directly highlights the expressive power and potential of phase representations in time series modeling. Nevertheless, there are a few exceptions. For example, on the relatively simple and highly periodic ETTh2 dataset, FITS slightly outperforms PhaseFormer. This suggests that for datasets with simpler structures and more predictable trends, some specialized baseline models may still exhibit better accuracy. It is also worth noting that TimeBase, which adopts a phase-based strategy, achieves competitive results on the relatively simple ETT datasets. In contrast, PhaseFormer demonstrates its advantage primarily on Traffic and Electricity, which are more complex and challenging datasets. This distinction illustrates that while phase-inspired models may be effective in straightforward settings, PhaseFormer generalizes better and excels in more demanding real-world contexts.

### A.3.2 THE DETAILED FORECASTING EFFICIENCY RESULTS

We further provide the efficiency comparison of PhaseFormer against all baselines in terms of FLOPs and number of parameters, with the input length set to 720 and the output length fixed at 96. The results in Fig. 5 reveal that PhaseFormer achieves a favorable trade-off between accuracy and efficiency. Despite its stronger predictive performance, PhaseFormer maintains moderate model size and computational cost, often comparable to or even lower than other transformer-based models: On complex datasets such as Traffic, PhaseFormer outperforms large baselines like PatchTST with substantially fewer FLOPs; On simpler datasets, even when specialized models such as TimeBase or FITS show competitive accuracy, their efficiency advantage diminishes when considering scalability to larger, real-world datasets. These findings underscore that PhaseFormer is not only accurate but also efficient, making it more suitable for deployment in resource-constrained or latency-sensitive environments.

Table 5: Parameters and FLOPS across models for different datasets.

| Model | Traffic | | Weather | | Electricity | | ETTh1 | | ETTh2 | | ETTm1 | | ETTm2 | |
|---|---|---|---|---|---|---|---|---|---|---|---|---|---|---|
| | Params | FLOPS | Params | FLOPS | Params | FLOPS | Params | FLOPS | Params | FLOPS | Params | FLOPS | Params | FLOPS |
| PhaseFormer | 1.156K | 13.9 | 308 | 0.15 | 1.156K | 5.18 | 1.156K | 0.11 | 1.156K | 0.11 | 1.156K | 0.11 | 1.156K | 0.11 |
| PatchTST | 7.589M | 498,577.49 | 1.373M | 1,054.77 | 1.373M | 16,122.93 | 587.68K | 51.29 | 587.68K | 51.29 | 587.68K | 51.29 | 587.68K | 51.29 |
| iTransformer | 6.731M | 11,652.34 | 5.153M | 257.54 | 5.153M | 3,347.97 | 369.9K | 8.12 | 304.1K | 6.68 | 304.1K | 7.29 | 304.1K | 7.29 |
| Crossformer | 22.954M | 259,209.90 | 158.34K | 84.09 | 13.537M | 96,564.63 | 2.069M | 544.20 | 2.069M | 544.20 | 2.069M | 544.20 | 2.069M | 544.20 |
| FEDformer | 21.206M | 13,679.70 | 5.828M | 2,757.24 | 11.861M | 6,904.61 | 5.792M | 2,734.51 | 5.792M | 2,734.51 | 5.793M | 2,734.51 | 5.793M | 2,734.95 |
| TimeBase | 214 | 8.44 | 214 | 0.21 | 214 | 3.14 | 214 | 0.07 | 214 | 0.07 | 704 | 0.23 | 704 | 0.23 |
| SparseTSF | 17.949K | 751.31 | 4.509K | 5.14 | 4.509K | 78.61 | 4.509K | 1.71 | 4.509K | 1.71 | 4.509K | 1.71 | 4.509K | 1.71 |
| FITS | 1.054K | 1.76 | 272 | 0.01 | 462 | 0.28 | 272 | 0.004 | 272 | 0.004 | 2.646K | 0.04 | 2.646K | 0.04 |
| TimeMixer | 5.697M | 2,026.53 | 5.562M | 205.40 | 5.584M | 739.64 | 4.024M | 125.91 | 4.024M | 125.91 | 4.024M | 125.95 | 4.024M | 125.95 |
| CycleNet | 563.216K | 720.27M | 421.928K | 17.55M | 472.328K | 268.22M | 419.576K | 5.85M | 419.576K | 5.85M | 419.576K | 5.85M | 419.576K | 5.85M |
| PatchMLP | 3.937M | 7,311.92M | 2.450M | 105.79M | 2.656M | 2,011.57M | 2.449M | 34.86M | 2.449M | 34.86M | 2.449M | 34.86M | 2.449M | 34.86M |

Table 6: Full results across datasets and prediction lengths. Each entry reports MAE and MSE. The input length is set to 720. The best results are marked with **Bold**, and the second-best results are marked with Underlined.

| Dataset | Horizon | PhaseFormer MSE | MAE | PatchTST MSE | MAE | iTransformer MSE | MAE | Crossformer MSE | MAE | FEDformer MSE | MAE | TimeBase MSE | MAE | SparseTSF MSE | MAE | FITS MSE | MAE | TimeMixer MSE | MAE | CycleNet MSE | MAE | PatchMLP MSE | MAE |
|---|---|---|---|---|---|---|---|---|---|---|---|---|---|---|---|---|---|---|---|---|---|---|---|
| ETTh1 | 96 | 0.359 | 0.382 | 0.377 | 0.408 | 0.389 | 0.421 | 0.408 | 0.442 | 0.485 | 0.500 | 0.365 | 0.387 | 0.362 | 0.389 | 0.380 | 0.402 | 0.410 | 0.441 | 0.379 | 0.403 | 0.417 | 0.44 |
| | 192 | 0.397 | 0.404 | 0.413 | 0.431 | 0.424 | 0.446 | 0.472 | 0.496 | 0.481 | 0.498 | 0.403 | 0.409 | 0.404 | 0.412 | 0.415 | 0.424 | 0.448 | 0.465 | 0.416 | 0.425 | 0.454 | 0.465 |
| | 336 | 0.425 | 0.424 | 0.436 | 0.444 | 0.456 | 0.469 | 0.480 | 0.486 | 0.522 | 0.521 | 0.409 | 0.419 | 0.435 | 0.426 | 0.449 | 0.460 | 0.475 | 0.490 | 0.447 | 0.445 | 0.514 | 0.503 |
| | 720 | 0.431 | 0.450 | 0.455 | 0.475 | 0.545 | 0.532 | 0.710 | 0.616 | 0.604 | 0.575 | 0.440 | 0.448 | 0.426 | 0.448 | 0.433 | 0.457 | 0.475 | 0.500 | 0.477 | 0.483 | 0.548 | 0.531 |
| ETTh2 | 96 | 0.275 | 0.338 | 0.276 | 0.339 | 0.305 | 0.361 | 1.164 | 0.744 | 0.401 | 0.451 | 0.292 | 0.350 | 0.294 | 0.346 | 0.271 | 0.336 | 0.315 | 0.380 | 0.271 | 0.337 | 0.324 | 0.382 |
| | 192 | 0.341 | 0.376 | 0.342 | 0.385 | 0.405 | 0.421 | 1.414 | 0.830 | 0.425 | 0.464 | 0.339 | 0.387 | 0.340 | 0.377 | 0.332 | 0.374 | 0.383 | 0.415 | 0.332 | 0.38 | 0.381 | 0.418 |
| | 336 | 0.369 | 0.405 | 0.364 | 0.405 | 0.411 | 0.436 | 1.220 | 0.794 | 0.427 | 0.471 | 0.394 | 0.420 | 0.360 | 0.398 | 0.355 | 0.396 | 0.415 | 0.436 | 0.362 | 0.408 | 0.408 | 0.44 |
| | 720 | 0.402 | 0.436 | 0.395 | 0.434 | 0.448 | 0.470 | 2.074 | 1.103 | 0.462 | 0.493 | 0.400 | 0.448 | 0.383 | 0.425 | 0.378 | 0.423 | 0.432 | 0.471 | 0.415 | 0.449 | 0.454 | 0.469 |
| ETTm1 | 96 | 0.293 | 0.344 | 0.298 | 0.352 | 0.315 | 0.369 | 0.306 | 0.353 | 0.406 | 0.441 | 0.311 | 0.351 | 0.314 | 0.359 | 0.313 | 0.357 | 0.332 | 0.384 | 0.307 | 0.353 | 0.319 | 0.367 |
| | 192 | 0.323 | 0.361 | 0.335 | 0.373 | 0.349 | 0.388 | 0.341 | 0.385 | 0.450 | 0.477 | 0.338 | 0.371 | 0.348 | 0.376 | 0.369 | 0.369 | 0.362 | 0.398 | 0.337 | 0.371 | 0.353 | 0.385 |
| | 336 | 0.358 | 0.381 | 0.366 | 0.389 | 0.381 | 0.409 | 0.383 | 0.420 | 0.436 | 0.466 | 0.364 | 0.386 | 0.368 | 0.386 | 0.367 | 0.385 | 0.386 | 0.413 | 0.364 | 0.387 | 0.381 | 0.401 |
| | 720 | 0.412 | 0.410 | 0.420 | 0.421 | 0.437 | 0.439 | 0.532 | 0.512 | 0.462 | 0.479 | 0.413 | 0.414 | 0.419 | 0.413 | 0.417 | 0.417 | 0.452 | 0.457 | 0.41 | 0.411 | 0.436 | 0.432 |
| ETTm2 | 96 | 0.163 | 0.256 | 0.165 | 0.260 | 0.179 | 0.274 | 0.244 | 0.338 | 0.339 | 0.406 | 0.162 | 0.256 | 0.167 | 0.259 | 0.166 | 0.256 | 0.192 | 0.285 | 0.159 | 0.249 | 0.186 | 0.272 |
| | 192 | 0.219 | 0.293 | 0.219 | 0.298 | 0.239 | 0.314 | 0.350 | 0.412 | 0.397 | 0.452 | 0.218 | 0.293 | 0.219 | 0.297 | 0.271 | 0.328 | 0.307 | 0.362 | 0.214 | 0.289 | 0.25 | 0.315 |
| | 336 | 0.269 | 0.326 | 0.268 | 0.333 | 0.309 | 0.356 | 0.400 | 0.431 | 0.418 | 0.452 | 0.270 | 0.328 | 0.271 | 0.330 | 0.352 | 0.380 | 0.380 | 0.412 | 0.268 | 0.326 | 0.298 | 0.347 |
| | 720 | 0.351 | 0.379 | 0.352 | 0.386 | 0.387 | 0.407 | 0.574 | 0.525 | 0.451 | 0.499 | 0.352 | 0.380 | 0.353 | 0.380 | 0.352 | 0.380 | 0.380 | 0.412 | 0.353 | 0.384 | 0.38 | 0.398 |
| Weather | 96 | 0.148 | 0.195 | 0.149 | 0.199 | 0.159 | 0.212 | 0.151 | 0.210 | 0.289 | 0.342 | 0.146 | 0.198 | 0.174 | 0.231 | 0.176 | 0.232 | 0.163 | 0.223 | 0.164 | 0.220 | 0.152 | 0.205 |
| | 192 | 0.193 | 0.237 | 0.193 | 0.243 | 0.203 | 0.252 | 0.220 | 0.273 | 0.340 | 0.403 | 0.185 | 0.241 | 0.216 | 0.267 | 0.203 | 0.256 | 0.201 | 0.255 | 0.209 | 0.258 | 0.197 | 0.247 |
| | 336 | 0.242 | 0.278 | 0.240 | 0.281 | 0.253 | 0.291 | 0.340 | 0.342 | 0.370 | 0.408 | 0.263 | 0.281 | 0.260 | 0.299 | 0.261 | 0.299 | 0.258 | 0.300 | 0.255 | 0.292 | 0.248 | 0.287 |
| | 720 | 0.309 | 0.332 | 0.312 | 0.334 | 0.317 | 0.337 | 0.362 | 0.393 | 0.420 | 0.421 | 0.314 | 0.331 | 0.325 | 0.345 | 0.325 | 0.346 | 0.329 | 0.348 | 0.320 | 0.338 | 0.324 | 0.340 |
| Electricity | 96 | 0.129 | 0.221 | 0.141 | 0.240 | 0.135 | 0.233 | 0.140 | 0.237 | 0.226 | 0.341 | 0.139 | 0.231 | 0.139 | 0.239 | 0.147 | 0.253 | 0.142 | 0.247 | 0.128 | 0.223 | 0.141 | 0.243 |
| | 192 | 0.148 | 0.238 | 0.156 | 0.256 | 0.155 | 0.253 | 0.165 | 0.259 | 0.220 | 0.336 | 0.153 | 0.245 | 0.155 | 0.250 | 0.159 | 0.256 | 0.164 | 0.273 | 0.143 | 0.237 | 0.163 | 0.265 |
| | 336 | 0.165 | 0.257 | 0.172 | 0.267 | 0.169 | 0.267 | 0.190 | 0.286 | 0.224 | 0.337 | 0.169 | 0.262 | 0.171 | 0.265 | 0.169 | 0.270 | 0.171 | 0.266 | 0.159 | 0.254 | 0.186 | 0.290 |
| | 720 | 0.201 | 0.285 | 0.208 | 0.299 | 0.204 | 0.301 | 0.227 | 0.312 | 0.271 | 0.378 | 0.207 | 0.294 | 0.208 | 0.300 | 0.214 | 0.302 | 0.209 | 0.313 | 0.197 | 0.287 | 0.221 | 0.320 |
| Traffic | 96 | 0.361 | 0.238 | 0.363 | 0.250 | 0.374 | 0.273 | 0.512 | 0.265 | 0.664 | 0.431 | 0.394 | 0.267 | 0.389 | 0.272 | 0.374 | 0.273 | 0.404 | 0.293 | 0.381 | 0.266 | 0.402 | 0.302 |
| | 192 | 0.373 | 0.243 | 0.382 | 0.258 | 0.393 | 0.283 | 0.528 | 0.271 | 0.613 | 0.382 | 0.407 | 0.270 | 0.399 | 0.272 | 0.393 | 0.282 | 0.404 | 0.292 | 0.394 | 0.273 | 0.419 | 0.309 |
| | 336 | 0.385 | 0.248 | 0.399 | 0.268 | 0.409 | 0.292 | 0.543 | 0.281 | 0.612 | 0.379 | 0.417 | 0.278 | 0.417 | 0.279 | 0.423 | 0.292 | 0.425 | 0.293 | 0.406 | 0.279 | 0.437 | 0.317 |
| | 720 | 0.428 | 0.270 | 0.432 | 0.289 | 0.450 | 0.314 | 0.598 | 0.314 | 0.664 | 0.410 | 0.456 | 0.298 | 0.449 | 0.299 | 0.450 | 0.314 | 0.453 | 0.314 | 0.441 | 0.300 | 0.479 | 0.342 |

### A.3.3 THE DETAILED RESULTS OF PCA VISUALIZATION

We present PCA visualization results on the ETTh1, ETTh2, ETTm1, ETTm2, Electricity, and Weather datasets in Fig. 8, in addition to Fig. 2b. The findings are consistent with those observed on the Traffic dataset: phase tokenization yields a significantly more compact space compared to patch tokenization.

### A.3.4 THE DETAILED RESULTS OF VARYING ROUTER NUMBERS

We further provide detailed results on the effect of varying the number of routers (1,2,4,8,16) across three datasets: Traffic, Electricity, and Weather. The input window was fixed at 720, and the output length was set to 96.

Our observations show that the number of routers does influence model performance, but the optimal configuration typically involves a relatively small number of routers. Specifically, the best performance was achieved with 8 routers on the Weather dataset, and with 4 routers on both the Electricity and Traffic datasets. Since routers serve as the foundation for aggregation and distribution in the phase token space, these results provide supporting evidence that the phase token space captures low-dimensional features, allowing strong performance even with fewer routers.

In practice, we recommend selecting the router number through a light-weight grid search, as the optimal value is typically small and stable across datasets. Consistent with our empirical findings, we observe that using routers within the range of 2–8 is generally sufficient to achieve strong and robust performance. This aligns with our analysis in Fig. 8 and Fig. 2b, where the phase representations exhibit a clear low-rank structure—fewer than eight principal components already explain over 90% of the total variance. Consequently, increasing the number of routers beyond this range yields only marginal improvements, and we do not observe any consistent trade-off patterns across datasets. Overall, these results suggest that the model does not require a large number of routers, and a modest choice of M can already provide a reliable and efficient configuration in most scenarios.

Table 7: Impact of router number $R$ on prediction accuracy. Each entry reports MSE and MAE. The input length is set to 720. The best results are marked with **Bold**, and the second-best results are marked with Underlined.

| Dataset | R=1 | | R=2 | | R=4 | | R=8 | | R=16 | |
|---|---|---|---|---|---|---|---|---|---|---|
| | MSE | MAE | MSE | MAE | MSE | MAE | MSE | MAE | MSE | MAE |
| Weather | 0.162 | 0.210 | 0.153 | 0.199 | 0.151 | 0.199 | **0.148** | **0.195** | 0.149 | 0.198 |
| Traffic | 0.372 | 0.249 | 0.367 | 0.243 | **0.361** | **0.238** | 0.364 | 0.242 | 0.368 | 0.243 |
| Electricity | 0.133 | 0.228 | 0.132 | 0.226 | **0.129** | **0.221** | 0.130 | 0.223 | 0.130 | 0.222 |

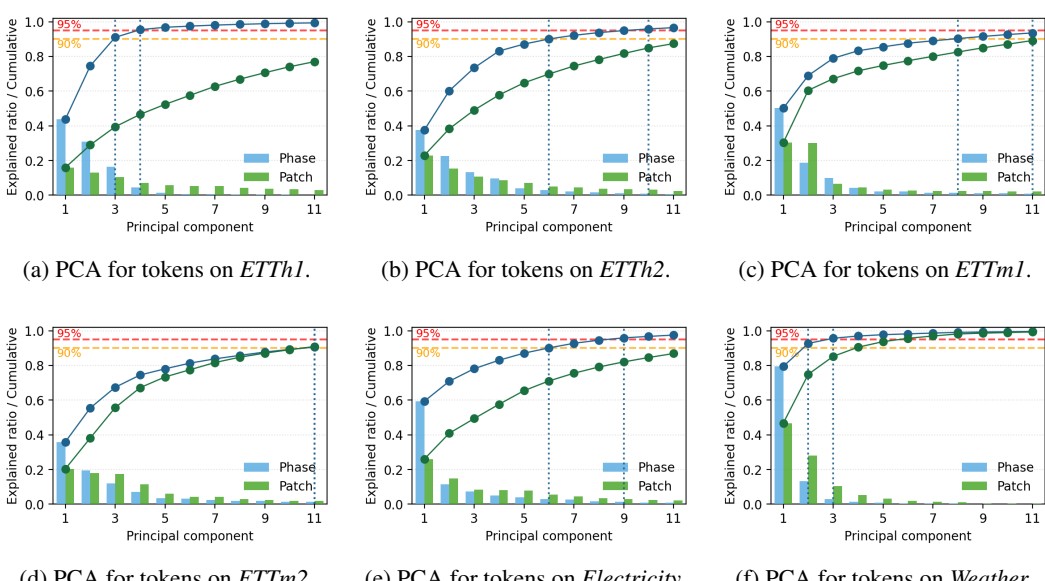

(a) PCA for tokens on *ETTh1*.  (b) PCA for tokens on *ETTh2*.  (c) PCA for tokens on *ETTm1*.

(d) PCA for tokens on *ETTm2*.  (e) PCA for tokens on *Electricity*.  (f) PCA for tokens on *Weather*.

Figure 8: Visualization of phase tokenization across six datasets: ETTh1, ETTh2, ETTm1, ETTm2, Electricity, and Weather.

### A.3.5 COMPARISON WITH TIME SERIES FOUNDATION MODELS

To contextualize the performance of our approach, we compare PhaseFormer against several representative foundation models drawn from the GIFT-Eval benchmark suite (Aksu et al., 2024). Specifically, we include Chronos (Ansari et al., 2024), Moirai (Woo et al., 2024), and the recently released Moirai-2 (Liu et al., 2025) as baseline models.

Due to space limitations, we report results on the Web/DevOps subsets of the benchmark in Tab. 8. These Web/DevOps datasets are *not* well-studied by many existing works (Nie et al., 2023; Zhang & Yan, 2023; Huang et al., 2025; Xu et al., 2024; Lin et al., 2024b). Their heterogeneous and highly non-stationary behaviors make them particularly challenging, and thus provide a realistic environment to evaluate the generalization capability of large time series foundation models.

Beyond demonstrating strong overall competitiveness, the empirical results clearly highlight the unique strengths of phase-based representation in handling *high-variance, noise-heavy workload patterns*. In particular, PhaseFormer consistently outperforms all foundation-model baselines on the `bitbrains_rnd` and `bitbrains_fast_storage` datasets—-two challenging subsets in GIFT-Eval due to frequent spikes and abrupt intensity shifts. These results indicate that Phase-Former's phase-aware design provides substantially improved robustness and stability under highly volatile real-world workload traces, where traditional architecture tend to degrade.

Table 8: Comparison of MSE (Mean) Across Four Forecasting Models on Web/CloudOps Datasets provided in GIFT-Eval.

| Dataset | PhaseFormer | Moirai-2 | Moirai-base | Chronos-base | Best Model |
|---|---|---|---|---|---|
| bizitobs_l2c/5T/short | 39.4347 | **20.6240** | 22.1000 | 24.1000 | Moirai-2 |
| bizitobs_l2c/H/short | 99.9582 | **67.8501** | 272.0000 | 243.0000 | Moirai-2 |
| bitbrains_rnd/H/short | **1,970,330** | 2,235,573 | 4,020,000 | 2,210,000 | PhaseFormer |
| bitbrains_rnd/5T/short | **1,606,034** | 1,660,532 | 180,000,000 | 1,800,000 | PhaseFormer |
| bizitobs_application/10S/short | 4,653,132 | **621,852** | 6,360,000 | 6,030,000 | Moirai-2 |
| bitbrains_fast_storage/H/short | **2,585,180** | 2,942,651 | 604,000,000 | 2,620,000 | PhaseFormer |
| bitbrains_fast_storage/5T/short | **1,887,398** | 2,142,125 | 25,400,000,000 | 1,920,000 | PhaseFormer |
| bizitobs_service/10S/short | 26,523 | **4,798** | 51,900 | 37,100 | Moirai-2 |

Table 9: Comparison of PhaseFormer variants. Each cell reports MSE, MAE, and FLOPs. The input length is fixed as 720 steps and the output length is fixed as 96 steps. The best results are marked with **Bold**, and the second-best results are marked with Underlined.

| Dataset | PhaseFormer-1.7K | | | PhaseFormer-5K | | | PhaseFormer-37K | | |
|---|---|---|---|---|---|---|---|---|---|
| | MSE | MAE | FLOPs | MSE | MAE | FLOPs | MSE | MAE | FLOPs |
| Electricity | **0.129** | **0.220** | **9.41M** | **0.129** | 0.221 | 31.97M | 0.131 | 0.223 | 221.05M |
| Traffic | 0.361 | 0.241 | **25.27M** | 0.366 | 0.243 | 85.84M | **0.360** | **0.236** | 593.61M |
| Weather | **0.150** | 0.199 | **0.62M** | 0.151 | **0.194** | 2.09M | 0.174 | 0.217 | 14.46M |

## A.4 ADDITIONAL HYPER PARAMETERS ANALYSIS

### A.4.1 IMPACT OF MODEL PARAMETER SCALE ON PERFORMANCE

We conducted comparative experiments on three variants of the PhaseFormer model with different parameter scales across the Electricity, Traffic, and Weather datasets. The three configurations are: a single-layer model with latent dimension 8 ($\approx$1.72K parameters), a single-layer model with latent dimension 16 ($\approx$5.48K parameters), and a two-layer model with latent dimension 32 ($\approx$37.1K parameters). Fig. 9 summarizes the results in terms of MSE, MAE and FLOPs.

Overall, the effect of model scale on performance is not consistent. On the Traffic dataset, larger models yield slight improvements, whereas on the Electricity and Weather datasets, the smaller and medium-sized models perform better. These findings indicate that PhaseFormer achieves a favorable balance between computational efficiency and predictive accuracy at relatively small parameter scales, and increasing model size does not lead to uniform gains across all tasks.

### A.4.2 IMPACT OF INPUT LENGTH ON PERFORMANCE

We examine how the input window size affects the prediction accuracy and computational cost of PhaseFormer. Throughout this section, $L$ denotes using the most recent $L$ time steps as model input. The output length is fixed as 96 steps. As summarized in Fig. 9, increasing $L$ reduces MSE and MAE across datasets, indicating that PhaseFormer benefits from longer historical context for modeling long-range temporal dependencies.

In terms of efficiency, the parameter count and FLOPs per forward pass remain nearly constant as $L$ increases, with only modest growth (see Fig. 10) attributable primarily to the embedding stage. This behavior arises because the sequence length processed by the core encoder/decoder is governed by the number of *phases*, which depends on the data's learned periodic structure rather than by the raw input length. Consequently, scaling $L$ mainly affects the embedding computations, whose cost is relatively small compared to the phase-based modules.

## A.5 ANALYSIS OF FINE-GRAINED TEMPORAL INFORMATION RETENTION

To examine whether Phase Tokenization preserves essential intra-cycle dynamics after compressing periodic structures, we conduct a reconstruction experiment on the Traffic dataset. Given an input sequence of length 720, PhaseFormer is trained to reconstruct sequences of the same length, removing forecasting uncertainty and isolating the representational capacity of the phase embeddings.

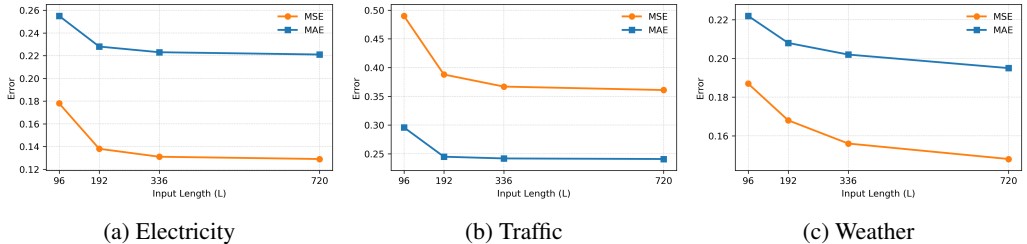

| (a) Electricity | (b) Traffic | (c) Weather |

Figure 9: Prediction error test results across datasets under different input lengths.

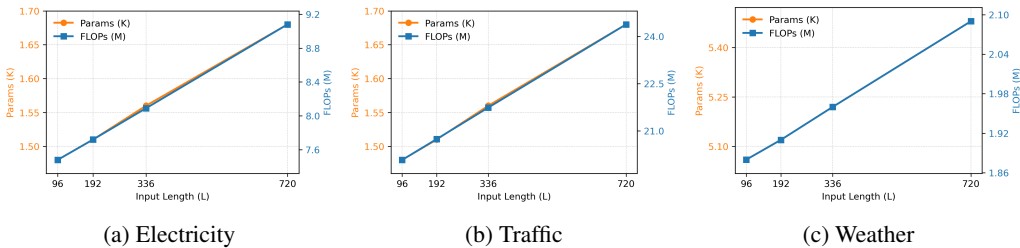

| (a) Electricity | (b) Traffic | (c) Weather |

Figure 10: Efficiency evaluation of PhaseFormer across datasets with varying input lengths.

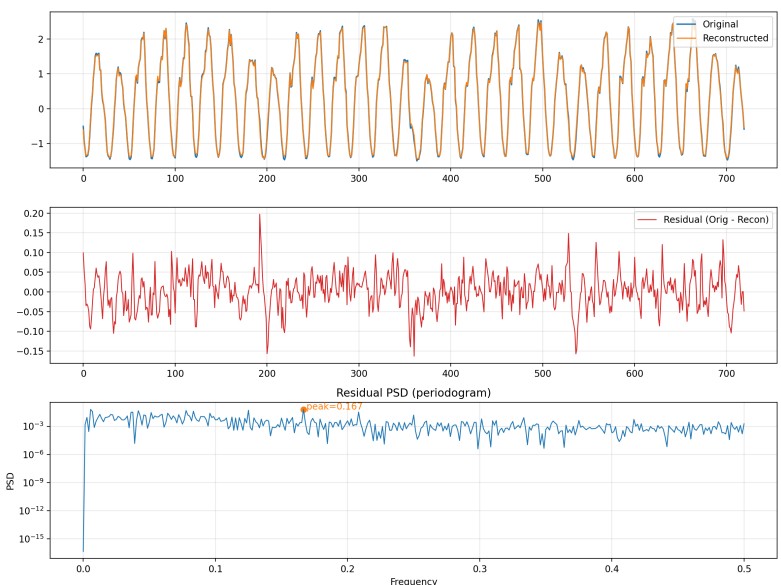

Figure 11: Time-domain and power-spectrum visualizations of the reconstruction residuals when PhaseFormer reconstructs 720 input time steps on the Traffic dataset.

**Setup.** We evaluate the retention of fine-grained information by computing the reconstruction residual and analyzing it in both the time and frequency domains. In particular, we estimate the power spectral density (PSD) of the residual to detect whether certain frequency components, especially those associated with short-term or high-frequency dynamics, are systematically diminished.

**Frequency Analysis of the Residual.** The PSD analysis in Fig. 11 shows that residual energy is evenly distributed across the spectrum, with no noticeable concentration or suppression in any frequency band. This indicates that Phase Tokenization does not introduce frequency-dependent information loss. Despite compressing each cycle into a compact phase representation, the model retains the ability to capture fine-grained intra-cycle dynamics, supporting the effectiveness of phase-based temporal abstraction for long-sequence forecasting.

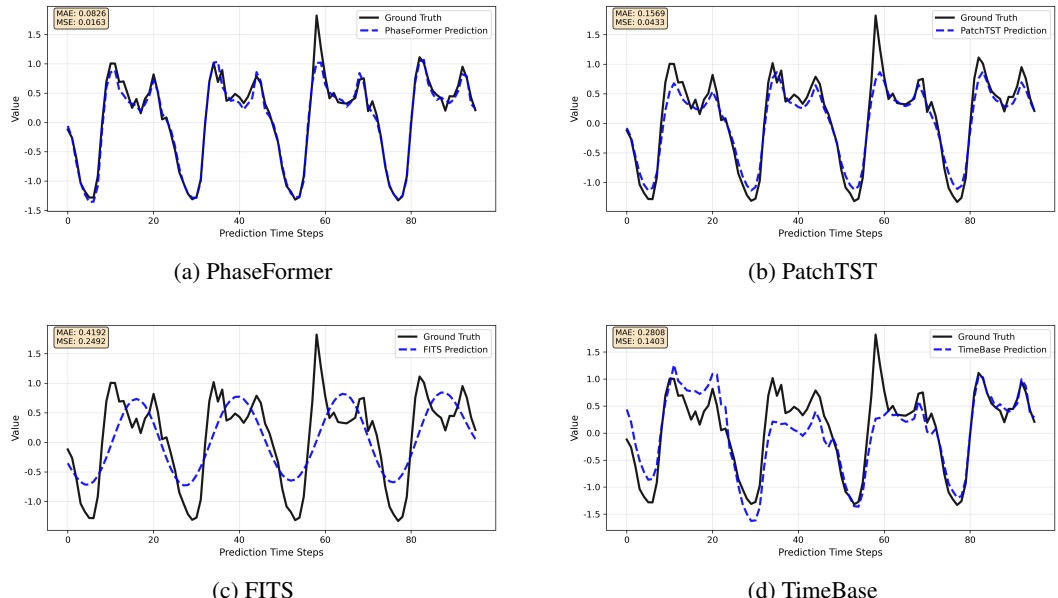

Figure 12: Visualization of forecasting results on Traffic dataset. The black lines stand for the ground truth and the blue lines stand for forecasting results.

## A.6 SHOWCASES

To provide a clearer comparison of predictive performance across different models, we present the results of PhaseFormer, PatchTST, FITS, and TimeBase on the Traffic dataset. PhaseFormer demonstrates strong predictive performance, as reflected by both the shape of its forecasts and the actual prediction errors.

Compared with PhaseFormer, PatchTST produces lower peak values within each cycle, failing to fully match the true curve. This discrepancy is likely due to phase shifts in the periodic pattern that reduce peak amplitudes. FITS, which performs prediction in the frequency domain with frequency band partitioning, tends to overlook high-frequency information. As a result, its predictions deviate from the ground truth, though the forecasts still preserve a periodic structure. TimeBase aligns well with the general cyclical pattern but fails to capture true variations across cycles, a limitation stemming from its patch-based basis construction mechanism.

## A.7 THEORETICAL ANALYSIS FOR PHASE TOKENIZATION

Under the classical finite-order linear system assumption, the structural dependencies in a time series (especially periodic or quasi-periodic components) are reflected in the rank properties of suitable delay-embedding matrices.

Consider a univariate time series

$$x_1, x_2, \ldots, x_T.$$

Fix a collection of delays (lags) $\tau_1, \ldots, \tau_D$ and a set of reference indices $s_1, \ldots, s_H$ such that $s_h + \tau_D \leq T$ for all $h$. The corresponding $D \times H$ delay-embedding matrix is

$$X = \begin{bmatrix} x_{s_1+\tau_1} & x_{s_2+\tau_1} & \cdots & x_{s_H+\tau_1} \\ x_{s_1+\tau_2} & x_{s_2+\tau_2} & \cdots & x_{s_H+\tau_2} \\ \vdots & \vdots & & \vdots \\ x_{s_1+\tau_D} & x_{s_2+\tau_D} & \cdots & x_{s_H+\tau_D} \end{bmatrix}.$$

The classical Hankel matrix is recovered as the special case with unit delay and consecutive indices, e.g. $\tau_d = d - 1$ and $s_h = h$, but our analysis does not rely on this specific structure.

If the series is dominated by periodic or harmonic components, it can often be generated (or well approximated) by a low-dimensional linear dynamical system. In this case, the delay-embedding

matrix $X$ exhibits a natural **low-rank** structure: its columns lie approximately in a low-dimensional subspace determined by the underlying system dynamics. Only when the series resembles unstructured white noise does the delay-embedding matrix approach full rank.

For a univariate periodic (or quasi-periodic) series, such a delay-embedding admits the factorization

$$X = AG,$$

where $A$ and $G$ encode, respectively, the delay-dependent responses and the coefficients associated with different reference indices (e.g., days or windows). The parameters $D$, the choice of delays $\{\tau_d\}$, and the number of columns $H$ jointly determine the shape of the embedding.

Thus, we model the data matrix as

$$X = AG^\top + N \in \mathbb{R}^{D \times H},$$

where $A \in \mathbb{R}^{D \times r}$, $G \in \mathbb{R}^{H \times r}$ are column full rank with $\text{rank}(A) = \text{rank}(G) = r$, and $N$ is noise. The true signal is $M = AG^\top$. We assume $r \ll \min(D, H)$.

Patch tokenization corresponds to the row space $\text{Row}(X)$ (the right singular $r$-subspace), while phase tokenization corresponds to the column space $\text{Col}(X)$ (the left singular $r$-subspace).

A shared transformation applies $S \in \mathbb{R}^{H \times H}$ on the hourly dimension:

$$X' = XS^\top = A(SG)^\top + N'.$$

In our formulation, the set $S$ represents a family of linear transformations applied to the input sequence. Such transformations frequently arise in practical systems due to sensor delays, timing jitters, or preprocessing procedures including smoothing or temporal alignment.

For a matrix $Y$, define the spectral separation as

$$\text{sep}_r(Y) := \min_{i \le r, j > r} |\sigma_i(Y) - \sigma_j(Y)|.$$

When $\text{rank}(Y) = r$, this equals $\sigma_r(Y)$. In particular,

$$\delta = \sigma_r(M), \quad \delta' = \sigma_r(MS^\top), \quad \delta_{\min} = \min(\delta, \delta').$$

Assume $S$ is invertible on $\text{Col}(G)$, i.e. $\text{rank}(SG) = r$. Define

$$\kappa := \sigma_{\min}(S|_{\text{Col}(G)}) > 0.$$

Moreover, since $\sigma_r(M) \ge \sigma_r(A)\sigma_r(G)$ and $\sigma_r(MS^\top) \ge \kappa\sigma_r(A)\sigma_r(G)$, we have the useful bound

$$\delta_{\min} \ge \min(1, \kappa)\, \sigma_r(A)\sigma_r(G).$$

For two $r$-dimensional subspaces $\mathcal{U}, \mathcal{V}$, their distance is

$$d(\mathcal{U}, \mathcal{V}) := \|P_\mathcal{U} - P_\mathcal{V}\|_2 = \sin\Theta_{\max}(\mathcal{U}, \mathcal{V}),$$

where $P_\mathcal{U}$ is the orthogonal projector onto $\mathcal{U}$. This metric satisfies the triangle inequality.

**Lemma 1 (Column space preservation)** *Let* $M = AG^\top$. *If* $\text{rank}(SG) = r$, *then*

$$\text{Col}(MS^\top) = \text{Col}(M) = \text{Col}(A).$$

*If* $\text{rank}(SG) < r$, *then the column space shrinks.*

**Lemma 2 (Row space change)** *For* $M = AG^\top$,

$$\text{Row}(M) = \text{Col}(G), \qquad \text{Row}(MS^\top) = \text{Col}(SG).$$

*As a result:*

$$d(\text{Col}(G), \text{Col}(SG)) > 0 \iff S(\text{Col}(G)) \ne \text{Col}(G).$$

**Lemma 3 (Wedin's** $\sin\Theta$ **theorem)** *Let* $\widehat{M} = M + E$ *and* $\delta = \text{sep}_r(M) > 0$. *Then*

$$d(\mathcal{U}_r(M), \mathcal{U}_r(\widehat{M})) \le C\frac{\|E\|_2}{\delta},$$

*where* $\mathcal{U}_r(M)$ *denotes the leading left singular $r$-subspace of $M$ (the right case is analogous). Here $C$ is an absolute constant (often $C \in [2, 2\sqrt{2}]$). The condition $\delta > 0$ is necessary.*

**Theorem 2** *Assume* $\mathrm{rank}(SG) = r$ *and* $\delta, \delta' > 0$. *Then:*

1. *For phase tokenization,*

$$d\big(\mathcal{U}_r(X), \mathcal{U}_r(X')\big) \;\leq\; C\Big(\tfrac{\|N\|_2}{\delta} + \tfrac{\|N'\|_2}{\delta'}\Big) \;\leq\; C\frac{\|N\|_2 + \|N'\|_2}{\delta_{\min}}.$$

*In the noiseless case, Lemma 1 ensures exact invariance, so the distance is* $0$.

2. *For patch tokenization,*

$$d\big(\mathcal{V}_r(X), \mathcal{V}_r(X')\big) \;\geq\; d_0 - C\Big(\tfrac{\|N\|_2}{\delta} + \tfrac{\|N'\|_2}{\delta'}\Big),$$

*where* $d_0 := d(\mathrm{Col}(G), \mathrm{Col}(SG))$. *If* $S(\mathrm{Col}(G)) \neq \mathrm{Col}(G)$, *then* $d_0 > 0$.

**Proof 1** *For phase tokenization, apply the triangle inequality:*

$$d(\mathcal{U}_r(X), \mathcal{U}_r(X')) \leq d(\mathcal{U}_r(X), \mathcal{U}_r(M)) + d(\mathcal{U}_r(M), \mathcal{U}_r(MS^\top)) + d(\mathcal{U}_r(MS^\top), \mathcal{U}_r(X')).$$

*By Lemma 1 the middle term vanishes, and the two boundary terms are bounded by Wedin's theorem, yielding the stated inequality.*

*For patch tokenization, we have*

$$d(\mathcal{V}_r(X), \mathcal{V}_r(X')) \geq d(\mathcal{V}_r(M), \mathcal{V}_r(MS^\top)) - d(\mathcal{V}_r(M), \mathcal{V}_r(X)) - d(\mathcal{V}_r(MS^\top), \mathcal{V}_r(X')).$$

*By Lemma 2 the first term equals* $d_0$, *and the other two are controlled by Wedin's theorem, proving the bound.*

In real-world scenarios, slight variations in timing or conditions occur from day to day, so the daily transformations are not exactly identical. We model this systematic inconsistency by introducing a small perturbation $\Delta_d$, which captures the mismatch between the ideal linear transformation $S$ and the actual data-generating process. Suppose each day's transform is $S_d = S + \Delta_d$ with $\|\Delta_d\|_2 \leq \varepsilon$. Then

$$X' = XS^\top + R, \qquad R_{d,:} = X_{d,:}\Delta_d^\top.$$

Bounding row by row gives $\|R_{d,:}\|_2 \leq \varepsilon \|X_{d,:}\|_2$, hence

$$\|R\|_F \leq \varepsilon \|X\|_F \quad \Rightarrow \quad \|R\|_2 \leq \varepsilon(\|M\|_F + \|N\|_F).$$

The stability of tokenization-induced subspaces measures whether the model is able to maintain consistent internal representations of key periodic or trend components when subjected to these perturbations. Correspondingly, the distances between tokenization-induced subspaces provide a quantitative assessment of how much the representation changes, with smaller distances indicating more stable and reliable token features.

Phase-based tokenization exhibits near-invariance under these transformations, offering an inductive bias that is particularly beneficial for real-world forecasting scenarios.

**Theorem 3 (Stability under day-wise perturbations)** *Under the relaxed model, each day uses* $S_d = S + \Delta_d$ *with* $\|\Delta_d\|_2 \leq \varepsilon$, *so that*

$$X' \;=\; XS^\top + R, \qquad R_{d,:} = X_{d,:}\Delta_d^\top.$$

*Let* $X = M+N$ *with* $M = AG^\top$, $\mathrm{rank}(A) = \mathrm{rank}(G) = r$, *and assume* $\mathrm{rank}(SG) = r$ *so that* $\delta = \sigma_r(M) > 0$ *and* $\delta' = \sigma_r(MS^\top) > 0$. *Define* $\delta_{\min} = \min(\delta, \delta')$ *and* $d_0 = d(\mathrm{Col}(G), \mathrm{Col}(SG))$. *Then there exists an absolute constant* $C \in [2, 2\sqrt{2}]$ *such that:*

1. **Phase tokenization (left** $r$**-subspace):**

$$d\big(\mathcal{U}_r(X), \mathcal{U}_r(X')\big) \;\leq\; C\left(\frac{\|N\|_2}{\delta} + \frac{\|N'\|_2}{\delta'} + \frac{\|R\|_2}{\delta'}\right)$$

$$\leq\; C\,\frac{\varepsilon(\|M\|_F + \|N\|_F) + \|N\|_2 + \|N'\|_2}{\delta_{\min}}.$$

2. **Patch tokenization (right $r$-subspace):**

$$d\big(\mathcal{V}_r(X), \mathcal{V}_r(X')\big) \;\geq\; d_0 \;-\; C\left(\frac{\|N\|_2}{\delta} + \frac{\|N'\|_2}{\delta'} + \frac{\|R\|_2}{\delta'}\right)$$

$$\geq\; d_0 \;-\; C\,\frac{\varepsilon(\|M\|_F + \|N\|_F) + \|N\|_2 + \|N'\|_2}{\delta_{\min}}.$$

In particular, if $S(\mathrm{Col}(G)) = \mathrm{Col}(G)$ then $d_0 = 0$ and patch tokenization is also preserved up to the same perturbation scale.

Moreover, using $\delta_{\min} \geq \min(1, \kappa)\,\sigma_r(A)\sigma_r(G)$ with $\kappa = \sigma_{\min}(S|_{\mathrm{Col}(G)}) > 0$ makes the role of signal strength explicit.

**Proof 2** *By row-wise control, $\|R_{d,:}\|_2 \leq \varepsilon\|X_{d,:}\|_2$, hence*

$$\|R\|_F \leq \varepsilon\|X\|_F \leq \varepsilon(\|M\|_F + \|N\|_F), \qquad \|R\|_2 \leq \|R\|_F \leq \varepsilon(\|M\|_F + \|N\|_F).$$

*Insert the chain*

$$X \;\to\; M \;\to\; MS^\top \;\to\; MS^\top + N' \;\to\; X' = MS^\top + N' + R.$$

*For phase subspace $\mathcal{U}_r$, according to the triangle inequality,*

$$d\big(\mathcal{U}_r(X), \mathcal{U}_r(X')\big) \leq d\big(\mathcal{U}_r(X), \mathcal{U}_r(M)\big) + d\big(\mathcal{U}_r(M), \mathcal{U}_r(MS^\top)\big)$$
$$+ d\big(\mathcal{U}_r(MS^\top), \mathcal{U}_r(MS^\top + N')\big) + d\big(\mathcal{U}_r(MS^\top + N'), \mathcal{U}_r(X')\big).$$

*The middle term vanishes by Column space preservation (Lemma 1). Applying Wedin's $\sin\Theta$ theorem (Lemma 3) to the remaining three perturbations $E \in \{N, \ N', \ R\}$ yields $C\|N\|_2/\delta + C\|N'\|_2/\delta' + C\|R\|_2/\delta'$. Use $\delta_{\min} \leq \delta, \delta'$ and Step 1 to obtain Item 1.*

*For patch subspace $\mathcal{V}_r$, we use the reverse triangle inequality:*

$$d\big(\mathcal{V}_r(X), \mathcal{V}_r(X')\big) \geq d\big(\mathcal{V}_r(M), \mathcal{V}_r(MS^\top)\big) - d\big(\mathcal{V}_r(M), \mathcal{V}_r(X)\big)$$
$$- d\big(\mathcal{V}_r(MS^\top), \mathcal{V}_r(MS^\top + N')\big) - d\big(\mathcal{V}_r(MS^\top + N'), \mathcal{V}_r(X')\big).$$

*The first term equals $d_0$ by Row space change (Lemma 2). Apply Wedin's theorem to the other three terms to obtain the final results.*

Therefore, we can conclude that our Phase Tokenization performs better than Patch Tokenization when facing periodic perturbations and is more tolerant of mild periodic fluctuations. The above theoretical analysis can be applied to real-world time-series data that have a stable primary period but exhibit small variations.

## A.8 THE USE OF LARGE LANGUAGE MODELS

In this work, large language models (specifically ChatGPT-5) are used solely for polishing the writing, identifying grammatical issues, and performing proofreading.

