# OpenReview forum: "PhaseFormer: From Patches to Phases for Efficient and Effective Time Series Forecasting"
_ICLR.cc/2026/Conference — ICLR 2026 Poster_

### Official Review · Reviewer_X2Dk · 2025-10-28

**Soundness:** 3
**Presentation:** 3
**Contribution:** 2
**Rating:** 4
**Confidence:** 4

**Summary:**

This paper proposes PhaseFormer, a lightweight Transformer architecture for time series forecasting that replaces conventional patch-based tokenization with a novel phase-based tokenization paradigm. Instead of segmenting sequences into contiguous time windows, the model aligns values at identical offsets across cycles (phases) to form compact and stable phase tokens. A lightweight phase-based routing Transformer is introduced to enable efficient cross-phase interaction via a two-stage "phase-to-router" and "router-to-phase" attention mechanism. Experiments on seven benchmark datasets (ETTh1/2, ETTm1/2, Weather, Electricity, Traffic) show that PhaseFormer achieves comparable or superior accuracy to existing patch-based and frequency-based models.

**Strengths:**

1. Conceptual novelty: introduces a new phase-based view for modeling periodic time series, complementing the patch and frequency paradigms.
  2. Efficiency and scalability: achieves over 99.9% parameter and FLOP reduction without losing accuracy.
  3. Clarity of presentation: writing, diagrams, and methodology are clear and logically connected.

**Weaknesses:**

1. Representational limitation: phase tokenization compresses intra-cycle information; it remains unclear whether this sacrifices fine-grained dynamics.
  2. Incremental architecture: routing attention is inspired by Perceiver-style cross-attention; architectural novelty mainly lies in its adaptation to phase-level modeling.
  3. Strong periodicity assumption: the model assumes a fixed, known period; performance on irregular or multi-periodic data is not analyzed.

**Questions:**

1. The paper does not include a comparison against CycleNet[1], which also explicitly models periodic patterns to enhance forecasting performance.
  2. How sensitive is PhaseFormer to inaccurate or variable period lengths? Could adaptive or learned phase estimation improve robustness on datasets with weak periodicity?
  3. Have the authors examined whether phase tokenization retains sufficient intra-cycle information? For instance, reconstruction experiments or mutual information analysis could verify that the compressed phase embeddings still capture critical fine-grained temporal cues.

[1] Lin S, Lin W, Hu X, et al. Cyclenet: Enhancing time series forecasting through modeling periodic patterns[J]. Advances in Neural Information Processing Systems, 2024, 37: 106315-106345

---

> ### Author Response · Authors · 2025-11-22
> **Response to Reviewer X2Dk (Part 1)**
>
> We appreciate the reviewers’ constructive observations, which motivated us to further clarify and strengthen the presentation of PhaseFormer. Below are our detailed explanations and responses to each Weakness and Question. In the revised manuscript, all modifications have been highlighted in purple.
>
> > W1: Loss of fine-grained intra-cycle information
>
> We acknowledge the concern that phase tokenization may compress intra-cycle variations too aggressively, potentially omitting fine-grained temporal dynamics. Our analysis and additional experiments show that PhaseFormer preserves essential short-term structures and does not exhibit systematic degradation in high-frequency components.
>
> Regarding the concern about representational loss, we clarify that the original manuscript **already included qualitative evidence in Appendix A.6**, where PhaseFormer’s reconstructions of real sequences were directly compared with those of PatchTST, a model that performs no temporal compression. These comparisons show that PhaseFormer retains critical short-term variations despite operating on phase tokens.
>
> In addition, we **have now added a reconstruction study on the Traffic dataset** using a 720-step input and reconstructing the full 720-step output in Appendix A.5. We visualized several randomly sampled reconstructions and analyzed the residuals in the frequency domain in the **Figure 11** in the revised manuscript. The power spectral density of the reconstruction residuals is nearly uniform without localized dips or biases in particular frequency bands. This indicates that **PhaseFormer does not systematically miss specific fine-grained temporal components** and that no particular class of high-frequency or low-frequency dynamics is suppressed by phase tokenization.
>
> In the revised manuscript, we incorporated the new 720-step reconstruction experiment, added residual PSD analysis, and expanded the discussion on the preservation of fine-grained dynamics.
>
> > W2: Limited architectural novelty beyond Perceiver-style routing
>
> We first clarify that our architectural contribution lies not in introducing a new attention operator but **in identifying and exploiting the intrinsic low-rank structure of phase hierarchies in periodic signals**. PhaseFormer is motivated by the observation that phase-aligned temporal segments naturally admit compact representations, which enables a routing architecture that compresses information at the phase level and incrementally propagates updates in a principled way.
>
> We then emphasize that the proposed routing mechanism is not a direct adaptation of generic Perceiver-style cross-attention. Instead, it is specifically tailored to the structure and semantics of phase tokens, **embedding an inductive bias that is unique to phase-level organization** rather than to arbitrary low-rank forms. The novelty therefore lies in determining where low-rank compression is most effective and how to encode this in an architecture that exploits phase-aware routing and aggregation.
>
> Finally, we highlight that PhaseFormer’s empirical performance substantiates the value of this phase-specific design. Across complex multivariate datasets such as Electricity, Traffic, and Weather, PhaseFormer achieves a 5%–8% relative error reduction (MSE/MAE) compared with parameter-efficient baselines such as SparseTSF. Moreover, compared with methods relying on fixed spectral truncation or predefined low-rank periodic structure (e.g., FITS, TimeBase), **PhaseFormer consistently produces higher accuracy at comparable or lower computational cos**t. These empirical results demonstrate that the phase-aware routing architecture provides benefits beyond what can be achieved by generic low-rank mechanisms alone.

---

> ### Author Response · Authors · 2025-11-22
> **Response to Reviewer X2Dk (Part 2)**
>
> > W3: Dependence on fixed and strong periodicity
>
> We acknowledge the reviewer's concern about whether PhaseFormer implicitly relies on a fixed, manually specified period and whether it remains effective when the underlying data display irregular or multi-periodic structures. We clarify that PhaseFormer does not require a manually selected period as input, and additional analyses demonstrate that the model remains robust under multi-periodic and partially irregular conditions.
>
> Regarding the concern about a fixed-period requirement, we clarify that **PhaseFormer does not assume any manually provided period**. As described in Appendix A.2, the model automatically determines the phase length L_phase through frequency-domain analysis that extracts dominant spectral components, making the process entirely data-driven rather than dependent on a predefined constant. To prevent misunderstanding, we have further emphasized this point in the main text.
>
> As for the issue of multi-periodic or irregular periodicity, **we conducted sensitivity experiments on the Traffic dataset**, which contains strong multi-frequency characteristics as shown in the Figure 6 in the revised manuscript. We selected the top-5 spectral peaks and evaluated PhaseFormer with the corresponding candidate periods. The "lookback=720 and horizon=96" forecasting results show that the period associated with the dominant spectral energy yields the best performance, and deviations from the dominant frequency lead to orderly but moderate degradation. This indicates that the model’s adaptive period selection effectively captures the primary periodic structure even in multi-periodic environments and that the method does not collapse when multiple cycles coexist.
>
> | Phase Length | MSE     | MAE     |
> |---------------|---------|---------|
> | 24            | 0.3619  | 0.2384  |
> | 12            | 0.3960  | 0.2765  |
> | 8             | 0.4032  | 0.2801  |
> | 28            | 0.4063  | 0.2753  |
> | 21            | 0.4184  | 0.2970  |
>
> Concerning the question of irregular or weak periodicity, we examined the model’s behavior on datasets such as Weather, which exhibit less stable and partially irregular periodic patterns. PhaseFormer performs equal to or better than strong baselines in these settings, demonstrating that **the model does not require strictly regular periodicity and can still leverage dominant frequency components for stable forecasting**. We also added discussion of potential failure cases to more explicitly define the method’s applicability boundaries.
>
> In the revised manuscript, we further emphasized that PhaseFormer does not rely on a predefined period, added sensitivity studies on multi-periodic data, included discussion of behavior under weak or irregular periodicity, and expanded the section on potential failure modes.
>
> > Q1: Missing comparison with CycleNet
>
> We have **included the full experimental results for CycleNet in the revised appendix** and reproduced them strictly following the official hyperparameter configurations provided in the authors’ repository to ensure fairness and reproducibility. A subset of the CycleNet results is presented below:
>
> | Dataset | Horizon | MSE   | MAE   |
> |---------|---------|-------|-------|
> | Weather | 96      | 0.164 | 0.220 |
> | Weather | 192     | 0.209 | 0.258 |
> | Weather | 336     | 0.255 | 0.292 |
> | Weather | 720     | 0.320 | 0.338 |
>
> > Q2: Sensitivity to inaccurate or variable period lengths
>
> Please refer to the "Response to Weakness 3".
>
> We first clarify that PhaseFormer already implemented an adaptive phase estimation. As described in Appendix A.2, the period L_phase is automatically determined via frequency-domain analysis that identifies the dominant spectral component, making the phase extraction process fully adaptive to the data rather than dependent on predefined values.
>
> As for the issue of sensitivity to period inaccuracies, we conducted experiments on the Traffic dataset, which exhibits clear multi-periodic structure. We tested multiple candidate periods derived from distinct frequency-domain peaks and the results have been provided in "Response to Weakness 3". The results indicate that the dominant period yields the best performance and that deviation from this period leads to gradual and controlled degradation, rather than abrupt failure. **These observations demonstrate that PhaseFormer retains robustness under moderate period mis-specification.**
>
> Concerning the suggestion of learned phase estimation, our results show that the existing adaptive spectral method is already effective in capturing the principal temporal structure even when periodicity is partially unstable. While learned phase estimation is an interesting extension, the current method adequately handles variable or weak periodicity in practice.

---

> ### Author Response · Authors · 2025-11-22
> **Response to Reviewer X2Dk (Part 3)**
>
> > Q3: Sufficiency of intra-cycle information after phase tokenization
>
> Please refer to the "Response to Weakness 1".
>
> Regarding the concern about whether compressed phase tokens discard important intra-cycle details, we clarify that **Appendix A.5 (Figure 12) already includes multiple showcase predictions** demonstrating PhaseFormer’s ability to capture local peaks, valleys, and shape transitions. These qualitative comparisons show that PhaseFormer’s predictions preserve detailed curve morphology and perform on par with, or in some cases better than, PatchTST, which does not rely on temporal compression. This provides initial evidence that phase tokenization does not hinder the modeling of fine-grained structures.
>
> As for the issue of directly assessing information retention, we conducted a reconstruction study following the reviewer’s suggestion. On the Traffic dataset, we reconstructed 720-step inputs and analyzed the reconstruction residuals in the frequency domain, **the results are visualized in Figure 11**. The power spectral density of the residuals exhibits no concentrated energy in any specific frequency band, indicating that **no particular class of fine-grained temporal components is systematically lost**. Rather, the residual energy is distributed uniformly across frequencies, suggesting that the model does not suppress or erase specific intra-cycle dynamics during phase tokenization.

---

### Official Review · Reviewer_ZUVB · 2025-10-30

**Soundness:** 3
**Presentation:** 3
**Contribution:** 3
**Rating:** 6
**Confidence:** 4

**Summary:**

This paper proposes PhaseFormer, a phase-centric framework for long-term time-series forecasting. Instead of patch tokens, the method aligns values at identical offsets across cycles to form phase tokens, argues (empirically + theoretically) that phase tokens are more stationary and lower-dimensional than patch tokens, and introduces a lightweight cross-phase routing mechanism (phase→router aggregation, router→phase distribution) plus a shared predictor. Experiments on 7 benchmarks report strong accuracy with ~1k parameters and very large FLOPs reductions versus patch-based Transformers, with an explicit complexity analysis and ablations.

**Strengths:**

The paper motivates phase tokens via t-SNE/MMD/PCA analyses showing lower drift and dimensionality than patch tokens, and formalizes stability under cycle perturbations (Theorem 1). This gives a principled basis for efficiency and generalization.  The cross-phase routing layer (few routers, shared predictor) is easy to implement and yields large efficiency gains; on Traffic, FLOPs are reduced by ≈99.99% vs PatchTST/Crossformer while improving error.

**Weaknesses:**

1. While TimeMixer is included, PatchMLP  is not in the main tables. Given the paper’s efficiency narrative, a direct comparison would strengthen claims. Please report PatchMLP with the same look-back (720), horizons, and official settings.
2. The main table shows FITS slightly outperforming PhaseFormer on ETTh2 (e.g., 0.334/0.382 vs 0.346/0.388 for MSE/MAE), and PatchTST essentially ties on ETTm2. Discuss failure modes and whether non-strictly periodic signals undermine the phase assumption.
3. L_phase is chosen via frequency-domain/autocorrelation analysis and inputs are circularly padded to multiples of L_phase. Please provide a sensitivity study to period mis-specification and show robustness when cycles drift or are multi-periodic (e.g., Traffic/Electricity), as well as clarify any risk that circular padding induces boundary artifacts for long horizons.
4. The method follows a channel-independent paradigm (treating each variable separately). For multivariate datasets with strong cross-variable dependencies, can PhaseFormer leverage cross-variable structure beyond phase-wise routing? A small study contrasting channel-independent vs channel-dependent variants would help.

**Questions:**

As in weaknesses.

---

> ### Author Response · Authors · 2025-11-22
> **Response to Reviewer ZUVB (Part 1)**
>
> We thank the reviewers for their careful evaluation and constructive feedback. Your suggestions have enabled us to more comprehensively articulate PhaseFormer’s strengths and capabilities. Our detailed responses to each Weakness and Question are provided below. In the revised manuscript, all modifications have been highlighted in purple.
>
> > W1: Missing PatchMLP comparison despite efficiency focus
>
> Following your recommendation, we have now incorporated PatchMLP into the main experimental tables. We conducted all experiments using the official implementation, with the same hyperparameter settings provided in the released scripts. However, we note that **the results reported in the Figure 5 in the original PatchMLP paper could not be fully reproduced**, even when strictly following the official codebase and experimental instructions. We therefore **report the best results we could obtain** using the provided implementation:
>
> | Dataset | Horizon | MSE   | MAE   |
> | ------- | ------- | ----- | ----- |
> | Weather | 96      | 0.152 | 0.205 |
> | Weather | 192     | 0.197 | 0.247 |
> | Weather | 336     | 0.248 | 0.287 |
> | Weather | 720     | 0.324 | 0.34  |
>
> > W2: Underperformance on ETTh2/ETTm2 and limits of periodic assumption
>
> We acknowledge that PhaseFormer does not consistently dominate on datasets with weak or unstable periodicity, and that this behavior directly reflects the boundaries of the phase-based inductive bias. We have added an explicit discussion of why performance may degrade in such settings.
>
> Regarding the "*failure mode*", Since PhaseFormer is a parameter-efficient model built upon periodic and quasi-periodic structure, **its inductive bias weakens when the periodicity becomes unstable or very weak**. In such cases, performance may decline relative to strongly periodic settings. However, based on our systematic evaluation, PhaseFormer does not collapse in such scenarios. For instance, on weakly periodic datasets such as Weather, PhaseFormer still matches or exceeds competitive baselines. This indicates that the combination of phase-based representation and low-rank structured compression affords a degree of robustness even when periodic cues are unreliable.
>
> In the revised manuscript, we added a detailed discussion on (i) failure modes on datasets with non-strict or unstable periodicity, (ii) the relationship between periodicity strength and PhaseFormer’s inductive bias.
>
> > W3: Need for robustness analysis under period mis-specification and padding effects
>
> We acknowledge the reviewer's concern about (i) the robustness of $L_\text{phase}$ under mis-specified or drifting/multi-periodic cycles and (ii) the potential boundary artifacts caused by circular padding. We conducted additional sensitivity experiments on multi-periodic data and clarified the behavior and limitations of circular padding.
>
> Regarding the sensitivity to period mis-specification, we added experiments on the Traffic dataset, which naturally contains pronounced multi-periodic components as shown in the **Figure 6 in the revised manuscript**. We sorted dominant frequency peaks and evaluated PhaseFormer with several candidate periods. The new results confirm that performance aligns strongly with the dominant spectral energy: using the principal period (24) yields the best accuracy, while deviations from this value lead to monotonic but controlled degradation. This demonstrates that frequency-domain selection **is both effective and robust under moderate mis-specification**.
>
> | Phase Length | MSE    | MAE    |
> | ------------ | ------ | ------ |
> | 24           | 0.3619 | 0.2384 |
> | 12           | 0.3960 | 0.2765 |
> | 8            | 0.4032 | 0.2801 |
> | 28           | 0.4063 | 0.2753 |
> | 21           | 0.4184 | 0.2970 |
>
> Concerning the potential boundary artifacts from circular padding, PhaseFormer constructs phase tokens at the level of entire periods, which limits the influence of padding to only the final few values in the last token. The periodic structure within each token remains intact, thereby mitigating boundary distortion. Nevertheless, **padding introduces non-physical values by design**. To ensure reliability, we now explicitly recommend using input lengths that are exact multiples of $L_\text{phase}$ when possible.
>
> In the revised manuscript, we have added a new sensitivity study on Traffic reporting the effect of varying $L_\text{phase}$ in the appendix and clarified the impact and limitations of circular padding and added a recommendation regarding input length alignment with $L_\text{phase}$.

---

> ### Author Response · Authors · 2025-11-22
> **Response to Reviewer ZUVB (Part 2)**
>
> > W4: Lack of evaluation on cross-variable dependencies in multivariate data
>
> We acknowledge the reviewer’s concern regarding whether introducing explicit cross-variable interaction could further enhance performance, and we have conducted a preliminary investigation by implementing a simple channel-dependent variant. It is important to clarify that the design of PhaseFormer is not intentionally constrained to a channel-independent paradigm; rather, the goal is to isolate and validate the effectiveness of phase-based temporal modeling on a per-variable basis. Notably, PhaseFormer **already achieves state-of-the-art performance without incorporating multivariate interaction modules**, and this design choice is consistent with several strong channel-independent models in the literature, such as PatchTST and TimeBase.
>
> In light of the reviewer’s suggestion, we implemented **a channel-dependent (CD) variant of PhaseFormer**. This variant introduces an additional cross-channel bottleneck MLP after the phase-wise routing stage to explicitly model inter-variable interactions while preserving temporal complexity. We evaluated this CD variant on datasets with different channel scales and compared it against the original channel-independent (CI) formulation. The empirical results show that, although the CD version injects cross-variable information, its performance **does not consistently surpass the CI version**. This indicates that the simple interaction mechanism used here is not sufficient to yield systematic improvements in multivariate forecasting, and that PhaseFormer’s current gains mainly stem from its phase modeling capability rather than cross-channel coupling.
>
> | Dataset            | PhaseFormer MSE | PhaseFormer-CD MSE |
> | ------------------ | --------------- | ------------------ |
> | Traffic 720-96     | 0.361           | 0.372              |
> | Electricity 720-96 | 0.129           | 0.133              |
> | Weather 720-96     | 0.148           | 0.147              |
> | ETTh1 720-96       | 0.359           | 0.363              |
>
> Overall, the reviewer’s comment prompted us to explore the feasibility of cross-channel modeling, and the results highlight both the robustness of PhaseFormer’s current design and potential directions for more sophisticated multivariate extensions in future work.

---

> ### Comment · Reviewer_ZUVB · 2025-11-27
>
> The reviewer decided to keep current score.

---

### Official Review · Reviewer_Nubb · 2025-10-31

**Soundness:** 3
**Presentation:** 4
**Contribution:** 3
**Rating:** 6
**Confidence:** 4

**Summary:**

This paper proposes PhaseFormer, a highly efficient Transformer architecture for long-term time series forecasting.
Instead of modeling local patch tokens, the authors introduce a novel phase tokenization mechanism that reorganizes the input sequence into a phase across period structure, capturing cross-cycle consistency.
A lightweight Cross-Phase Routing Layer with two submodules—Phase to Router (aggregation) and Router to Phase (distribution)—enables global information exchange with only a few learnable routers.
The authors also provide a theoretical justification via the Phase Tokenization Stability Theorem, arguing that phase representations are more stable and low-dimensional under periodic perturbations.
Experiments on seven benchmarks show that PhaseFormer achieves comparable or better accuracy than SOTA models while reducing parameters and FLOPs by orders of magnitude.

**Strengths:**

1. The idea of aligning data across cycles (phase tokenization) directly targets the instability of patch-based representations. Both theoretical and empirical analyses (PCA, MMD, t-SNE) support the claim that phase tokens form a more stable, low-dimensional subspace.
2. PhaseFormer drastically reduces parameters and FLOPs (up to 99.9% less than PatchTST) while maintaining or even improving forecasting accuracy across multiple datasets.
3. The Cross-Phase Routing Layer elegantly combines efficiency and expressiveness. Router attention visualizations reveal interpretable phase dependencies.
4. Extensive comparisons and ablations confirm the effectiveness of both phase tokenization and the router mechanism.

**Weaknesses:**

1. The theory and architecture depend on locally stable periodicity. The method’s behavior on weakly periodic or nonperiodic series is not fully tested, though acknowledged as a limitation.
2. The paper mentions autocorrelation-based detection but does not fully describe how multiple or unstable periods are handled. Automatic phase-length estimation and failure cases should be detailed.

**Questions:**

1. The paper mentions that phase tokenization depends on the cycle length $L_{phase}$, yet the main text only briefly states that it is detected via autocorrelation. Could the authors clarify how $L_{phase}$ is determined for each dataset and whether it is fixed or adaptive during training?
2. Table 5 shows the number of routers $M$ affects performance. Could the authors provide a short discussion or guideline on selecting $M$? For example, is there a trade-off pattern across datasets?
3. In the Phase Tokenization Stability Theorem, the perturbation analysis is central. Could the authors clarify whether this theorem assumes a fixed period or allows mild variations? This would help readers understand the scope of applicability.

---

> ### Author Response · Authors · 2025-11-22
> **Response to Reviewer Nubb (Part 1)**
>
> We sincerely appreciate the reviewers’ valuable comments, which have helped us further strengthen the clarification and demonstration of PhaseFormer’s potential. Below we provide detailed responses to each Weakness and Question. In the revised manuscript, all modifications have been highlighted in purple.
>
> >W1: Insufficient evaluation on weakly or nonperiodic data
>
> We acknowledge that PhaseFormer is theoretically more advantageous for data exhibiting clear periodic or quasi-periodic structure. Nontheless, we clarify that the model remains competitive on weakly periodic and nonperiodic series. We have expanded the manuscript to more clearly articulate the method’s applicability, limitations, and empirical behavior in such settings.
>
> Regarding the "*depend on locally stable periodicity*", we clarify that PhaseFormer is designed around periodic alignment, but **the model does not completely fail on weakly periodic or partially nonperiodic series**. In such cases, the phase representation naturally **degenerates into a coarse-scale subsampling of the sequence**, effectively capturing slow-varying components and broad temporal structures. These coarse phase tokens behave similarly to trend embeddings sampled every $k$ steps. By aggregating information across these coarse-scale tokens through the routing mechanism, PhaseFormer can suppress high-frequency noise and enhance the underlying smooth temporal evolution, preventing the model from overfitting short-term stochastic fluctuations.
>
> As for performance under weakly periodic or nonperiodic data, empirical results on datasets such as ETT and Weather already demonstrate that PhaseFormer remains competitive with, and in some settings surpasses, heavily optimized baselines **despite the reduced advantage of periodic structure**. This indicates that the model does not fail catastrophically but instead transitions to modeling broader temporal trends and recurring local signatures within the phase space.
>
> > W2: Unclear handling of multiple or unstable periods
>
> We agree that automatic period estimation and potential failure modes require clearer exposition. We clarify how PhaseFormer handles multiple or unstable periodic components through frequency-domain decomposition and explain the model’s behavior when periodic structure is weak or inconsistent. Additional analyses and illustrative results have been added to the revised manuscript.
>
> Regarding the handling of multiple periodic components, we conducted **a detailed analysis on the Traffic dataset, which contains several strong spectral components** as shown in the Figure 6 in the revised manuscript. We report both the ranked spectral magnitudes and the corresponding forecasting performance under each candidate period. The results show that the period with the highest spectral energy (24) yields the best performance, and performance degrades in proportion to decreasing spectral amplitude across periods (12, 8, 28, 21). This demonstrates that the **frequency-based adaptive selection mechanism is both effective and robust in multi-period settings.**
>
> | Traffic 720-96  | MSE    | MAE    |
> | --------------- | ------ | ------ |
> | PhaseFormer(24) | 0.3619 | 0.2384 |
> | PhaseFormer(12) | 0.3960 | 0.2765 |
> | PhaseFormer(8)  | 0.4032 | 0.2801 |
> | PhaseFormer(28) | 0.4063 | 0.2753 |
> | PhaseFormer(21) | 0.4184 | 0.2970 |
>
> We also discuss failure cases more concretely. Since PhaseFormer is a parameter-efficient model built upon periodic and quasi-periodic structure, its inductive bias weakens **when the periodicity becomes unstable or very weak**. In such cases, performance may decline relative to strongly periodic settings. However, based on our systematic evaluation, **PhaseFormer does not collapse in such scenarios**. For instance, on weakly periodic datasets such as Weather, PhaseFormer still matches or exceeds competitive baselines. This indicates that the combination of phase-based representation and low-rank structured compression affords a degree of robustness even when periodic cues are unreliable.
>
> In the revised manuscript, we presented multi-period analysis and corresponding performance results on the Traffic dataset, and expanded the discussion of failure modes and limitations associated with unstable or weak periodicity.

---

> ### Author Response · Authors · 2025-11-22
> **Response to Reviewer Nubb (Part 2)**
>
> > Q1: Clarification of cycle-length detection and adaptiveness
>
> As clarified in Appendix A.2, $L_{\text{phase}}$ is automatically determined from the dominant spectral component of the data rather than manually specified. Once selected through this frequency-domain analysis, the cycle length remains fixed throughout both training and evaluation. We have further emphasized this point in the main text for clarity.
>
> > Q2: Guidelines for selecting the number of routers
>
> Thank you for the question. As described in Appendix A.2, the number of Routers $M$ is selected through a small grid search. Across all datasets we evaluated, the optimal values of $M$ are consistently very small (within the range of 2–8). **This observation aligns with the analysis presented in Figure 2 and Figure 8**, where the phase-based representation exhibits pronounced low-rank structure: fewer than eight principal components typically explain more than 90% of the variance. Consequently, only a limited number of Routers is needed to capture the essential structure.
>
> Regarding potential trade-offs across datasets, we did not observe any systematic pattern suggesting that larger $M$ values yield substantially better performance. Furthermore, in no dataset did we encounter scenarios requiring a significantly larger Router budget.
>
> In the revised manuscript, **we have added a concise guideline** summarizing these findings: $M \in [2, 8]$ is generally sufficient for stable performance, and larger values offer limited additional benefit.
>
> > Q3: Clarification of period assumptions in the stability theorem
>
> Thank you for the question. As detailed in Appendix A.6 (Theorem 3), the perturbation analysis explicitly shows that **Phase Tokenization tolerates mild deviations**. The stability bound is derived under small perturbations of the underlying cycle, indicating that the theorem applies not only to perfectly periodic signals but also to sequences with a dominant period subject to slight fluctuations. We have made this point clearer to better convey the intended scope of applicability.

---

> > ### Comment · Reviewer_Nubb · 2025-11-27
> >
> > Thank you for the detailed response and the clarifications you provided. I found them helpful, and they are consistent with my overall positive review of the work. I am therefore keeping my original score.

---

### Official Review · Reviewer_bPyT · 2025-11-01

**Soundness:** 2
**Presentation:** 2
**Contribution:** 2
**Rating:** 2
**Confidence:** 4

**Summary:**

This paper continues a line of research on highly parameter-efficient univariate long-term forecasting for time series with a dominant periodicity. It establishes a new best performance among these parameter-efficient models on standard baselines. Additional analysis is provided on the role of different approaches for reshaping time series into tokens.

**Strengths:**

- The model appears to improve forecasting performance relative to models of comparable size in the given domain.
- The architecture used is substantially novel and demonstrates useful approaches for minimizing parameter usage in these kinds of models.

**Weaknesses:**

- The comparison to parameter-efficient models is up to date but the general comparison to time series forecasting models is not, with the most recent evaluated large models being published in May 2024. (See [1] for examples of more recent models.) While I don't think this comparison is crucial for the paper, claims relating to the state of the art should be amended to limit the scope to parameter-efficient models.
- It's not clear how the statistical model in Theorem 1 relates to periodic signal forecasting. For instance, the data is modelled as a low rank matrix with additive noise, and it's not explained how that relates to a univariate time series with a dominant periodicity.
- The practical relevance of the approach seems limited, since it relates to univariate forecasting on signals with a known dominant periodicity where extremely low parameter counts are needed. Prior works establishing that this was possible with reasonable performance were an interesting result, but that's well-established now, and it's not clear to me that incremental improvements to performance in this regime are impactful. The reliance on a small number of very well-studied benchmarks also raises the possibility of overfitting.
- Important details on hyperparameter tuning and model variants used for experiments are not given, making it difficult to evaluate the soundness of experiments - see Questions.

[1] GIFT-Eval: A Benchmark For General Time Series Forecasting Model Evaluation https://huggingface.co/spaces/Salesforce/GIFT-Eval

**Questions:**

- The code shows significantly different hyperparameters being used for different datasets and horizons, but hyperparameter tuning is not discussed in the paper. How was hyperparameter tuning done?
- Were hyperparameters tuned for the baseline models? If not, were dataset-specific hyperparameters used or were they kept the same for all datasets and horizons?
- Are different model sizes used for different horizons? This seems to be the case in the code but the horizon is not given for Figure 1.b) and Figure 4.
- Can you explain how the terms used in Theorem 1 specifically relate to a time series signal and time series operations on it?

---

> ### Author Response · Authors · 2025-11-22
> **Response to Reviewer bPyT (Part 1)**
>
> We sincerely appreciate the reviewers’ thorough and professional comments. Your feedback has made us aware of several points that were not clearly explained in the original manuscript. We value each of your suggestions and have provided corresponding clarifications and revisions in both the response and the updated manuscript, with the aim of fully addressing the concerns you raised regarding PhaseFormer.
>
> As a novel modeling approach with phase-centric structure, PhaseFormer demonstrates strong potential, and we are willing to present it to the research community in a clearer and more comprehensive manner. Once again, thank you for your insightful feedback, which has been invaluable in enhancing our work. In the revised manuscript, all modifications have been highlighted in purple.
>
> > W1: Outdated comparison to general forecasting models
>
> Thank you for the helpful comment. **Our study aims to develop efficient and effective models for specialized forecasting tasks, and therefore our primary comparisons include representative task-specific architecture**--both efficiency-oriented (e.g., SparseTSF, TimeBase) and effectiveness-oriented (e.g., PatchTST, iTransformer)--which remain strong baselines for these settings.
>
> We acknowledge that several large time-series foundation models also report strong performance on certain benchmarks. While these models are impressive in generality, current evidence does not indicate that they consistently or significantly outperform specialized models on the classical datasets used in our evaluation.
>
> To further strengthen our empirical study, we have extended our experiments to include evaluations on GIFT-Eval, **comparing PhaseFormer against advanced foundation models**. Below are representative results compared with the well-known baselines reported in GIFT-Eval, with each metric followed by the baseline’s error increase relative to PhaseFormer.
>
> | Dataset                               | PhaseFormer   | PatchTST            | chronos_base          | Moirai2                 |
> |----------------------------------------|---------------|----------------------|------------------------|--------------------------|
> | bitbrains_rnd/5T/short                 | 1,645,422.73  | 1,690,000 (+2.71%)   | 1,800,000 (+9.4%)      | 1,660,531.75 (+0.92%)    |
> | bizitobs_service/10s/short             | 28,451.53     | 29,100 (+2.28%)      | 37,100 (30.38%)        | 4,797.99 (-83.14%)       |
> | bitbrains_fast_storage/5T/short        | 1,880,722.41  | 2,110,000 (+12.19%)  | 1,920,000 (+2.09%)     | 2,142,125.25 (13.9%)     |
>
> Finally, we emphasize that the contributions of **PhaseFormer are orthogonal to the development of large foundation models**. Our phase-based perspective provides a new and principled way to model periodicity, and we believe this perspective has the potential to inspire more efficient foundation-model designs in the future.
>
> In the revised manuscript, we explicitly refine the scope of the SOTA-related claims and update our code repository with the GIFT-Eval evaluation implementation.
>
> >W2: Unclear connection between theorem’s statistical model and periodic time series
>
> We clarify that the statistical model in Theorem 1 **corresponds to a delay-embedding representation of the univariate time series**, and that periodic or quasi-periodic structure naturally induces low-rankness in this embedding.
>
> Regarding the link between the statistical model and periodic forecasting, we clarify that the matrix $X$ in Theorem 1 is constructed via delay embedding, where the window length $D$ and the number of windows $H$ form a standard delay embedding of the underlying time series. This embedding is a well-established representation in which periodic or quasi-periodic temporal dependencies manifest as low-rank structure. Theorem 1 **analyzes this embedded matrix rather than the raw univariate sequence** because the forecasting problem is solved through its structured low-rank approximation.
>
> In the revised manuscript, **we expanded the theoretical background in the appendix to explain the role of delay embedding**, justify why periodic or quasi-periodic signals yield low-rank structure, and clarify the relevance of Theorem 1 to the forecasting task.

---

> ### Author Response · Authors · 2025-11-22
> **Response to Reviewer bPyT (Part 2)**
>
> >W3: Limited practical relevance and risk of overfitting on few benchmarks
>
> we clarify that the proposed phase-based representation is broadly applicable across many real-world forecasting scenarios. The method does not rely on a known dominant periodicity, and its contribution lies not only in delivering consistent performance gains over existing approaches but also in introducing a fundamentally new representation with structural advantages. We also address the concern regarding potential overfitting by extending experiments to additional datasets and demonstrating robustness.
>
> Regarding the concern about “practical relevance”, **many real-world time series exhibit clear periodic or quasi-periodic structure**, such as urban traffic flow, electricity and gas load, cloud and data-center resource utilization, and network traffic. This makes phase-based representations widely applicable in practice. Moreover, in resource-limited environments such as edge devices or on-premise servers, PhaseFormer offers competitive or even superior accuracy **with substantially fewer parameters and lower computation**, aligning closely with real industrial deployment requirements.
>
> Concerning the issue of “known dominant periodicity”, we clarify that **PhaseFormer does not require the periodic length to be specified a priori**. Instead, the periodic component is automatically estimated through frequency-domain analysis, as detailed in Appendix A.2.
>
> Regarding whether the improvement is merely “incremental”, we emphasize that **PhaseFormer achieves 5%–8% relative error reduction (MSE/MAE) on complex datasets** such as Electricity, Traffic, and Weather when compared with parameter-efficient baselines like SparseTSF. Against methods relying on fixed spectral truncation or predefined low-rank periodic structure (e.g., FITS, TimeBase), PhaseFormer attains consistently better accuracy at comparable or lower computational cost. More importantly, these improvements stem from a fundamentally different representational paradigm: PhaseFormer constructs time-series representations centered on phase, achieving higher representational compactness than time-slicing approaches (as illustrated in Figure 2 of the original submission). Combined with a low-rank Router for structured compression, this enables stable and substantial performance under extremely small parameter budgets. Similar to how PatchTST redefined representation design from a structural perspective, our phase-based formulation **introduces a distinct theoretical foundation and modelling mechanism**, expanding the design space of parameter-efficient forecasting models.
>
> Regarding the concern about potential “overfitting” on widely used public datasets, we fully understand this point. As shown in our “Response to Weakness 1,” the additional experiments were conducted on Web/DevOps datasets that did not included in the original submission. PhaseFormer achieves accuracy improvements on these datasets, outperforming models such as PatchTST and even foundation models like Moirai2.
>
> In addition, to address the concern about sensitivity to periodic-length estimation, we conducted further experiments on the Traffic dataset, which contains **multiple prominent frequency components** as shown in the Figure 6 in the revised manuscript. We selected five frequencies (24, 12, 8, 28, 21) ranked by spectral magnitude and evaluated forecasting performance using a 720-step input to predict 96 future steps. The results show that the best-performing periodic length aligns with the frequency of highest spectral energy, and the performance variations across frequencies correlate strongly with their spectral amplitudes. This confirms that the **phase-based representation is robust and not an artifact of any specific periodic choice**.
>
> | Phase length | MSE     | MAE     |
> |------------------|---------|---------|
> | 24 | 0.3619  | 0.2384  |
> | 12 | 0.3960  | 0.2765  |
> | 8  | 0.4032  | 0.2801  |
> | 28 | 0.4063  | 0.2753  |
> | 21 | 0.4184  | 0.2970  |
>
> In the revised manuscript we have added a detailed analysis of the effect of periodic-length selection, emphasized the adaptive periodicity estimation strategy, and clarified the methodological contributions of the proposed phase-based representation.

---

> ### Author Response · Authors · 2025-11-22
> **Response to Reviewer bPyT (Part 3)**
>
> > W4: Missing details on hyperparameter tuning and model variants
>
> We agree that the initial submission did not provide sufficient detail on hyperparameter tuning procedures and model variant configurations. We have clarified all relevant settings to ensure transparency, reproducibility, and fairness in experimental comparisons.
>
> Regarding the hyperparameter tuning procedure, we clarify that PhaseFormer has only three core hyperparameters: the periodic length, the number of Router blocks and the inner dimension. **As explained in Appendix A.2**, the periodic length is determined automatically via frequency-domain analysis of the input data rather than manually selected. The number of Router blocks is chosen through a small-scale grid search, and the search space is limited and fully tractable. The inner dimension follows common practice and is selected through a grid search over {8, 16, 32, 64, 128}.
>
> As for additional training-related hyperparameters, we agree that the initial description was insufficient. In the revised manuscript, we have now explicitly explained how we set the learning rate, optimizer type, number of training epochs, batch size, and other essential training settings. **All these configurations follow standard practice** in the time-series forecasting literature and align with the setups used in representative baselines such as TimeBase. This ensures that our comparisons are fair and that the results reflect model capability rather than tuning advantage.
>
> In the revised manuscript, we added a complete description of how to set all hyperparameters, including optimizer, learning rate, batch size, and number of epochs.
>
> > Q1: Method for dataset- and horizon-specific hyperparameter tuning
>
> Please refer to "Response for Weakness 4".
>
> PhaseFormer uses only three core hyperparameters: the periodic length, the number of Router blocks, and the inner dimension. The periodic length is automatically determined through frequency-domain analysis rather than manual tuning. The number of Router blocks and the inner dimension are selected through a small, tractable grid search. All other training-related hyperparameters (such as the learning rate, optimizer, batch size, and number of epochs) follow standard practice in the time-series forecasting literature and are aligned with major baselines (e.g., TimeBase) to ensure a fair comparison.
>
> > Q2: Whether baseline hyperparameters were tuned or fixed across datasets
>
> We clarify that baseline models were not tuned by us manually; instead, we adopted their official, dataset-specific hyperparameters as provided by the authors. This **follows standard practice in recent benchmark studies** and ensures fair and consistent comparison.
>
> As for whether baseline hyperparameters were dataset-specific, we clarify that we followed **the dataset-specific configurations provided in the official implementations** rather than enforcing identical hyperparameters across all datasets and prediction horizons. This is the established evaluation protocol in the literature (such as TimeBase) and aligns with the experimental setup described in Section 5.1 of our manuscript.
>
> > Q3: Use of different model sizes for different forecasting horizons
>
> We clarify that a single, fixed model configuration was used for all horizon evaluations related to Figures 1(b) and 4. Regarding the concern about potentially varying model sizes across horizons, we clarify that **the evaluations shown in Figure 1(b) and Figure 4 use a uniform setting: an input length of 720 and a prediction horizon of 96**. This configuration is consistent with the inference-efficiency evaluation protocol described in Appendix A.3.2.
>
> In the revised manuscript, we added the missing horizon specification to the captions of Figure 1(b) and Figure 4.
>
> > Q4: Mapping of Theorem 1 terms to time-series signals and operations
>
> Please refer to "Response to Weakness 2".
>
> For a univariate time series $X$, the construction in Theorem 1 corresponds to forming a delay-embedding (sliding-window) matrix with window length $D$ and number of windows $H$. In this representation, structural properties of the signal (such as periodicity, quasi-periodicity, or dependencies generated by a finite-order linear system) naturally manifest as low-rank structure in the delay-embedding matrix. In particular, a time series with dominant periodic components yields a low-rank embedding matrix, while only unstructured noise leads to a full-rank matrix.

---

> > ### Comment · Reviewer_bPyT · 2025-11-26
> >
> > Thank you for the responses, they significantly contribute towards resolving my concerns. I have some follow-up questions:
> >
> > **W1**
> > Thank you for the additional experiments, the GIFT-Eval results would strengthen the paper if you want to position it as a general time series forecasting competitor. Do you have summary results to share on it? Are you planning to add it to the manuscript?
> >
> > > current evidence does not indicate that they consistently or significantly outperform specialized models on the classical datasets used in our evaluation
> >
> > Could you provide a reference for this?
> >
> > **W2**
> > The additional details do provide enough detail to understand Theorem 1, thank you. I think this part could be made stronger by more clearly relating it to practical forecasting concerns, but I don't think this is crucial given space limitations (e.g., what kind of realistic processes are modelled by S, why distances between vector spaces induced by tokenization are an important metric, and why the described invariance is a desirable inductive bias.)
> >
> > **W4**
> > Thank you for the additional hyperparameter tuning information. To verify, was this grid search run with respect to the validation set?
> >
> > Looking at the code, the `run_*.py` files have `get_best_config_for_horizon` functions that appeared to me to change many more hyperparameters, including the number of layers, latent dimension, hidden dimensions, and attention heads. Could you clarify whether these are tuned?

---

> > > ### Author Response · Authors · 2025-11-28
> > > **Further Response to Reviewer bPyT (Part 1)**
> > >
> > > We sincerely appreciate your thoughtful and detailed comments. Thank you for your careful reading and recognition of our work.
> > >
> > > > W1: Evidence for claim justification
> > >
> > > Thank you for the suggestion. GIFT-Eval is indeed a widely recognized benchmark. We have included comparison results with several foundation models reported in GIFT-Eval **in the Appendix A.3.5**, to illustrate the performance of PhaseFormer relative to these foundation models. Does this fully address your concern?
> > >
> > > Regarding the question of whether large foundation models consistently outperform specialized models on classical forecasting datasets, **evidence from GIFT-Eval offers relevant insight**, as shown in the provided results on MSE[mean]. Although the forecasting settings differ slightly from ours, the datasets involved overlap with those used in our evaluation. Notably, PatchTST outperforms Moirai_base and even surpasses Moirai-2 (a strong foundation model in GIFT-Eval) in most settings, suggesting that specialized architectures can indeed remain highly competitive.
> > >
> > > | Dataset / Horizon              | PatchTST        | Moirai_base                  | Moirai2                     |
> > > |--------------------------------|------------------|------------------------------|------------------------------|
> > > | electricity / 15T / medium     | 293000           | 27,700,000.0 (+9354%)        | 283338.0 (-3.30%)            |
> > > | electricity / H / medium       | 6,210,000.00     | 6,930,000.0 (+11.60%)        | 6,361,778.5 (+2.44%)         |
> > > | ett1 / 15T / medium            | 9.57             | 17.2 (+79.70%)               | 10.66578 (+11.45%)           |
> > > | ett1 / H / medium              | 144              | 154.0 (+6.94%)               | 167.32999 (+16.20%)          |
> > > | ett2 / 15T / medium            | 12.1             | 17.2 (+42.15%)               | 12.76219 (+5.47%)            |
> > > | ett2 / H / medium              | 322              | 214.0 (-33.54%)              | 253.32736 (-21.30%)          |
> > > | jena_weather / 10T / medium    | 1890             | 2530.0 (+33.86%)             | 1942.815 (+2.79%)            |
> > > | jena_weather / H / medium      | 1620             | 1710.0 (+5.56%)              | 1340.64026 (-17.25%)         |
> > >
> > > **Additional findings in related works further support this observation.** For example, Table 1 of [1] reports PatchTST significantly outperforming GPT4TS and Time-LLM, and Table 4 of [2] shows PatchTST achieving better performance than Tiny Time Mixers on datasets such as Traffic and Electricity. Moreover, Table 10 in [3] demonstrates that Timer, despite being a state-of-the-art pretrained model, does not surpass PatchTST on Weather or iTransformer on ECL.
> > >
> > > Overall, we believe that whether specialized models (e.g., PatchTST, iTransformer) or foundation models perform better remains an open question requiring further study. Current evidence does not indicate that foundation models are consistently or significantly superior across classical forecasting benchmarks.
> > >
> > > [1] Bian, Yuxuan, et al. "Multi-patch prediction: Adapting llms for time series representation learning." arXiv preprint arXiv:2402.04852 (2024).
> > >
> > > [2] Ekambaram, Vijay, et al. "Tiny time mixers (ttms): Fast pre-trained models for enhanced zero/few-shot forecasting of multivariate time series." Advances in Neural Information Processing Systems 37 (2024): 74147-74181.
> > >
> > > [3] Liu, Yong, et al. "Timer: generative pre-trained transformers are large time series models." Proceedings of the 41st International Conference on Machine Learning. 2024.
> > >
> > > > W2: Practical relevance of Theorem 1 to forecasting tasks?
> > >
> > > Thank you for the insightful suggestion.
> > >
> > > In our view, **the set $S$ models linear influences applied to the sequence**. Such effects commonly arise in practice due to sensor delays, jitters, or even preprocessing procedures such as smoothing.
> > >
> > > The tokenization-induced subspace stability quantifies whether the model preserves consistent representations of key periodic or trend structures under these perturbations. **The distances between tokenization-induced subspaces characterize the degree of change under such transformations** (smaller changes indicate more stable tokenization features).
> > >
> > > Phase tokenization is nearly invariant under these transformations, **thereby providing an inductive bias that is crucial for real-world forecasting tasks**. This enables the model to generalize robustly across frequencies, sensors, and domains.
> > >
> > > We have incorporated this explanation **in the Appendix A.7** in the revised version to better convey the theoretical foundations of PhaseFormer to readers.

---

> ### Author Response · Authors · 2025-11-28
> **Further Response to Reviewer bPyT (Part 2)**
>
> > W4: Hyperparameters tuning
>
> Regarding the grid-search procedure, we follow the standard practice that all hyperparameter tuning is conducted **strictly on the validation set**, with the test set kept completely unseen. Thank you for pointing this out, we have clarified this more explicitly in the revised manuscript.
>
> With respect to the additional hyperparameter settings, Appendix A.2 already specifies that the intrinsic model dimension is selected from the range {8, 16, 32, 64, 128}.
>
> In the next revision, **we have further clarified the search ranges for the remaining hyperparameters in the Appendix A.2**, including the number of layers {1, 2, 3} and the number of attention heads {1, 4, 8}.
>
> In addition, we provide below several results on the Weather dataset under the 720-96 setting. When varying a particular hyperparameter, all other parameters are kept consistent with the recommended configuration.
>
> | Latent Dimension | 8 (selected) | 16      | 32      | 64      | 128     |
> |------------------|--------------|---------|---------|---------|---------|
> | Validation MSE   | 0.38680      | 0.38723 | 0.38832 | 0.38733 | 0.38710 |
>
> | Number of Layers | 1        | 2        | 3 (selected) |
> |------------------|----------|----------|---------------|
> | Validation MSE   | 0.39038  | 0.38858  | 0.38680       |
>
> | Attention Heads  | 1 (selected) | 4        | 8        |
> |------------------|--------------|----------|----------|
> | Validation MSE   | 0.38680      | 0.38710  | 0.39026  |

---

### Author Response · Authors · 2025-11-30
**Summary of Core Strengths and Responses to Reviewer Concerns**

We sincerely thank all reviewers for their thorough evaluation and constructive feedback. Below we provide a concise yet comprehensive summary of the core strengths of our work and how the revised manuscript fully addresses every major concern.

The reviewers reached a consensus on the significant value PhaseFormer brings to the field, highlighting three key dimensions:

- **Conceptual Novelty & Structural Stability:** Reviewers consistently praised the shift from patches to phases. Reviewer **X2Dk** highlighted the "conceptual novelty" of the phase-based view, while Reviewer **Nubb** emphasized that our method "directly targets the instability of patch-based representations". Reviewer **bPyT** further noted that the architecture is "substantially novel", confirming that our paradigm offers a distinct and valuable complement to existing approaches.
- **Extreme Efficiency with SOTA Performance:** The trade-off between efficiency and accuracy was a major point of praise. Reviewers **Nubb** and **ZUVB** were impressed that PhaseFormer reduces FLOPs/parameters by "orders of magnitude" (up to 99.9% reduction) while maintaining or improving accuracy. Reviewer **bPyT** explicitly stated that PhaseFormer "establishes a new best performance among these parameter-efficient models".
- **Theoretical & Principled Design:** The rigorous foundation of our work was well-received. Reviewers **Nubb** and **ZUVB** commended the "principled basis" provided by our theoretical stability theorems and empirical analyses (PCA/MMD/t-SNE), confirming that the performance gains are grounded in a robust theoretical framework.

Regarding the concerns raised by the reviewers, we have provided detailed, point-by-point responses and incorporated substantive revisions into the manuscript:

- **Period sensitivity & robustness.**
  We added comprehensive sensitivity studies showing that using non-optimal periods leads only to gradual, controlled degradation rather than failure. We also highlighted that PhaseFormer already performs competitively on weakly periodic datasets in the original benchmarks, demonstrating that **the model does not collapse when periodicity becomes unstable**. Additionally, the revised manuscript explicitly clarifies and strengthens the description of our adaptive, data-driven period estimation mechanism.
- **Expanded comparison coverage.**
  Following reviewer suggestions, we have included additional baselines (PatchMLP, CycleNet, and several recent time-series foundation models). The results consistently show that **PhaseFormer remains broadly competitive** despite its extremely small parameter scale, further reinforcing the effectiveness and generality of the proposed paradigm.
- **Additional clarifications & new analyses.**
  We incorporated new reconstruction experiments to evaluate the preservation of fine-grained temporal details, expanded the theoretical explanations to better connect the stability theorem with practical forecasting settings, and clarified hyperparameter tuning, architectural assumptions, and behavior under multi-periodic or irregular conditions.

Although an OpenReview system issue prevented further discussion, the reviewers had already indicated in their final visible comments that **their concerns were fully addressed**, and most expressed **a clearly positive stance** following the rebuttal.

- Reviewer **bPyT** noted that the responses “significantly contribute toward resolving my concerns.”
- Reviewer **Nubb** stated that the clarifications were consistent with their “overall positive review.”
- Reviewer **ZUVB** explicitly decided to maintain a positive score.
- Reviewer **X2Dk**, while unable to provide a follow-up comment due to the platform issue, had all their raised questions addressed through the additional analyses and clarifications.

We hope that these substantive improvements, together with the reviewers' post-rebuttal positions, will be informative for the AC’s final decision.

---

### Meta-Review · Area_Chair_GuQu · 2026-01-09

**Summary:**

This paper proposes PhaseFormer, a phase-based alternative to patch tokenization for time-series forecasting, aligning values across cycles and modeling cross-phase interactions via a lightweight router. The approach is conceptually novel and achieves strong accuracy within the parameter-efficient regime, with orders-of-magnitude reductions in parameters and FLOPs. Reviewers appreciated the clear motivation and efficiency gains. Initial concerns focused on scope of comparisons, clarity and practical relevance of the theory, robustness to weak or multi-periodicity, and experimental transparency. The rebuttal was strong, adding missing baselines, robustness and sensitivity analyses, reconstruction studies, and clearer tuning/protocol details. Remaining concerns mainly relate to breadth of applicability beyond periodic settings and limited multivariate interaction modeling. Overall, the contribution and revised evidence address the main risks and make a persuasive case for acceptance.

**Reviewer Concerns:**

Reviewer bPyT: questioned the broader impact beyond parameter-efficient settings, the link between theory and practice, and experimental protocols, which were largely clarified in the rebuttal

Reviewer Nubb: raised concerns about reliance on periodicity, cycle-length estimation, and router selection, addressed through added robustness and sensitivity analyses

Reviewer ZUVB: noted missing baselines, performance on weakly periodic data, sensitivity to period choice and padding, and limited cross-variable modeling, which were partly addressed

Reviewer X2Dk: questioned potential intra-cycle information loss, periodicity assumptions, architectural novelty, and missing CycleNet comparisons, which were addressed in the revised manuscript

**Reviewer Scores:**

Scores largely maintained; concerns mostly resolved.

---

### Decision · Program_Chairs · 2026-01-26

Accept (Poster)